# HiGen: Hierarchical Graph Generative Networks

**Mahdi Karami**
`mahdi.karami@ualberta.ca`

## Abstract

Most real-world graphs exhibit a hierarchical structure, which is often overlooked by existing graph generation methods. To address this limitation, we propose a novel graph generative network that captures the hierarchical nature of graphs and successively generates the graph sub-structures in a coarse-to-fine fashion. At each level of hierarchy, this model generates communities in parallel, followed by the prediction of cross-edges between communities using separate neural networks. This modular approach enables scalable graph generation for large and complex graphs. Moreover, we model the output distribution of edges in the hierarchical graph with a multinomial distribution and derive a recursive factorization for this distribution. This enables us to generate community graphs with integer-valued edge weights in an autoregressive manner. Empirical studies demonstrate the effectiveness and scalability of our proposed generative model, achieving state-of-the-art performance in terms of graph quality across various benchmark datasets. Code available at `https://github.com/Karami-m/HiGen_main`.

## 1 Introduction

Graphs play a fundamental role in representing relationships and are widely applicable in various domains. The task of generating graphs from data holds immense value for diverse applications but also poses significant challenges (Dai et al., 2020). Some of the applications include: the exploration of novel molecular and chemical structures (Jin et al., 2020), document generation (Blei et al., 2003), circuit design (Mirhoseini et al., 2021), the analysis and synthesis of realistic data networks, as well as the synthesis of scene graphs in computer (Manolis Savva et al., 2019; Ramakrishnan et al., 2021).

In all the aforementioned domains, a common observation is the presence of locally heterogeneous edge distributions in the graph representing the system, leading to the formation of clusters or communities and hierarchical structures. These clusters represent groups of nodes characterized by a high density of edges within the group and a comparatively lower density of edges connecting the group with the rest of the graph. In a hierarchical structure that arise from graph clustering, the communities in the lower levels capture the local structures and relationships within the graph. These communities provide insights into the fine-grained interactions among nodes. On the other hand, the higher levels of the hierarchy reflect the broader interactions between communities and characterize global properties of the graph. Therefore, in order to generate realistic graphs, it is essential for graph generation models to learn this multi-scale structure, and be able to capture the cross-level relations. While hierarchical multi-resolution generative models were developed for specific data types such as voice (Oord et al., 2016), image (Reed et al., 2017; Karami et al., 2019) and molecular motifs (Jin et al., 2020), these methods rely on domain-specific priors that are not applicable to general graphs with unordered nature. To the best of our knowledge, there exists no data-driven generative models specifically designed for generic graphs that can effectively incorporate hierarchical structure.

Graph generative models have been extensively studied in the literature. Classical methods based on random graph theory, such as those proposed in Erdos & Rényi (1960) and Barabási & Albert (1999), can only capture a limited set of hand-engineered graph statistics. Leskovec et al. (2010) leveraged the Kronecker product of matrices but the resulting generative model is very limited in modeling the underlying graph distributions. With recent advances in graph neural networks, a variety of deep neural network models have been introduced that are based on variational autoencoders (VAE) (Kingma & Welling, 2013) or generative adversarial networks (GAN) (Goodfellow et al., 2020).

Some examples of such models include (De Cao & Kipf, 2018; Simonovsky & Komodakis, 2018; Kipf & Welling, 2016; Ma et al., 2018; Liu et al., 2019; Bojchevski et al., 2018; Yang et al., 2019) The major challenge in VAE based models is that they rely on heuristics to solve a graph matching problem for aligning the VAE's input and sampled output, limiting them to small graphs. On the other hand, GAN-based methods circumvent the need for graph matching by using a permutation invariant discriminator. However, they can still suffer from convergence issues and have difficulty capturing complex dependencies in graph structures for moderate to large graphs (Li et al., 2018; Martinkus et al., 2022). To address these limitations, (Martinkus et al., 2022) recently proposed using spectral conditioning to enhance the expressivity of GAN models in capturing global graph properties.

On the other hand, autoregressive models approach graph generation as a sequential decision-making process. Following this paradigm, Li et al. (2018) proposed a generative model based on GNN but it has high complexity of $\mathcal{O}(mn^2)$. In a distinct approach, GraphRNN (You et al., 2018) modeled graph generation with a two-stage RNN architecture for generating new nodes and their links, respectively. However, traversing all elements of the adjacency matrix in a predefined order results in $\mathcal{O}(n^2)$ time complexity making it non-scalable to large graphs. In contrast, GRAN (Liao et al., 2019) employs a graph attention network and generates the adjacency matrix row by row, resulting in a $\mathcal{O}(n)$ complexity sequential generation process. To improve the scalability of generative models, Dai et al. (2020) proposed an algorithm for sparse graphs that decreases the training complexity to $\mathcal{O}(\log n)$, but at the expense of increasing the generation time complexity to $\mathcal{O}((n+m)\log n)$. Despite their improvement in capturing complex statistics of the graphs, autoregressive models highly rely on an appropriate node ordering and do not take into account the community structures of the graphs. Additionally, due to their recursive nature, they are not fully parallelizable.

A new family of diffusion model for graphs has emerged recently. Continuous denoising diffusion was developed by Jo et al. (2022), which adds Gaussian noise to the graph adjacency matrix and node features during the diffusion process. However, since continuous noise destroys the sparsity and structural properties of the graph, discrete denoising diffusion models have been developed as a solution in (Haefeli et al., 2022; Vignac et al., 2022; Kong et al., 2023). These models progressively edit graphs by adding or removing edges in the diffusion process, and then denoising graph neural networks are trained to reverse the diffusion process. While the denoising diffusion models can offer promising results, their main drawback is the requirement of a long chain of reverse diffusion, which can result in relatively slow sampling. To address this limitation, Chen et al. (2023) introduced a diffusion-based graph generative model. In this model, a discrete diffusion process randomly removes edges while a denoising model is trained to inverse this process, therefore it only focuses on a portion of nodes in the graph at each denoising step.

In this work, we introduce HiGen, a **Hi**erarchical **G**raph **Gen**erative Network to address the limitations of existing generative models by incorporating community structures and cross-level interactions. This approach involves generating graphs in a coarse-to-fine manner, where graph generation at each level is conditioned on a higher level (lower resolution) graph. The generation of communities at lower levels is performed in parallel, followed by the prediction of cross-edges between communities using a separate graph neural network. This parallelized approach enables high scalability. To capture hierarchical relations, our model allows each node at a given level to depend not only on its neighbouring nodes but also on its corresponding super-node at the higher level. We address the generation of integer-valued edge weights of the hierarchical structure by modeling the output distribution of edges using a multinomial distribution. We show that multinomial distribution can be factorized successively, enabling the autoregressive generation of each community. This property makes the proposed architecture well-suited for generating graphs with integer-valued edge weights. Furthermore, by breaking down the graph generation process into the generation of multiple small partitions that are conditionally independent of each other, HiGen reduces its sensitivity to a predefined initial ordering of nodes.

## 2 BACKGROUND

A graph $\mathcal{G} = (\mathcal{V}, \mathcal{E})$ is a collection of nodes (vertices) $\mathcal{V}$ and edges $\mathcal{E}$ with corresponding sizes $n = |\mathcal{V}|$ and $m = |\mathcal{E}|$ and an adjacency matrix $\mathbf{A}^{\pi}$ for the node ordering $\pi$. The node set can be partitioned into $c$ communities (a.k.a. cluster or modules) using a graph partitioning function $\mathcal{F}: \mathcal{V} \rightarrow \{1, ..., c\}$, where each cluster of nodes forms a sub-graph denoted by $\mathcal{C}_i = (\mathcal{V}(\mathcal{C}_i), \mathcal{E}(\mathcal{C}_i))$

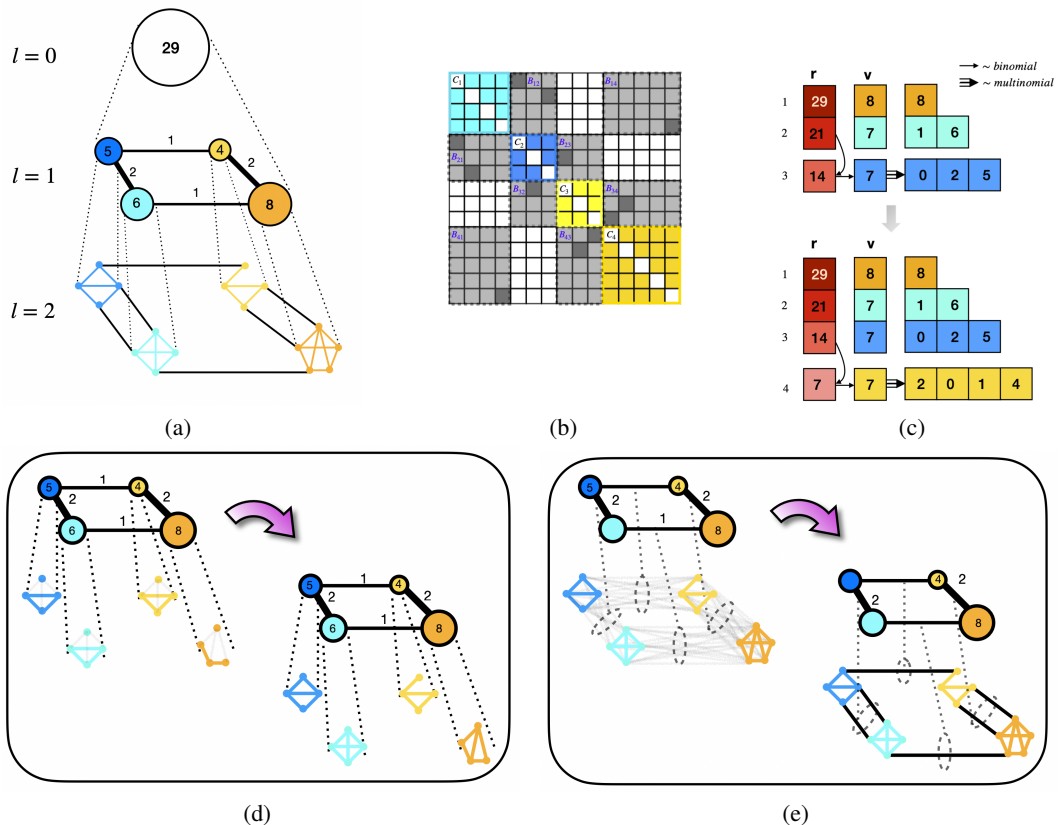

Figure 1: (a) A sample hierarchical graph, $\mathcal{HG}$ with 2 levels is shown. Communities are shown in different colors and the weight of a node and the weight of an edge in a higher level, represent the sum of the edges in the corresponding community and bipartite, respectively. Node size and edge width indicate their weights. (b) The matrix shows corresponding adjacency matrix of the graph at the leaf level, $\mathcal{G}^2$, where each of its sub-graphs corresponds to a block in the adjacency matrix, communities correspond to diagonal blocks and are shown in different colors while bipartites are colored in gray. (c) Decomposition of multinomial distribution as a recursive *stick-breaking* process where at each iteration, first a fraction of the remaining weights $r_t$ is allocated to the $t$-th row (corresponding to the $t$-th node in the sub-graph) and then this fraction, $v_t$, is distributed among that row of lower triangular adjacency matrix, $\hat{A}$. (d), (e) Parallel generation of communities and bipartites, respectively. Shadowed lines are the *augmented edges* representing candidate edges at each step.

with adjacency matrix $\mathbf{A}_i$. The cross-links between neighboring communities form a *bipartite graph*, denoted by $\mathcal{B}_{ij} = (\mathcal{V}(\mathcal{C}_i), \mathcal{V}(\mathcal{C}_j), \mathcal{E}(\mathcal{B}_{ij}))$ with adjacency matrix $\mathbf{A}_{ij}$. Each community is aggregated to a super-node and each bipartite corresponds to a super-edge linking neighboring communities, which induces a coarser graph at the higher (a.k.a. parent) level. Herein, the levels are indexed by superscripts. Formally, each community at level $l$, $\mathcal{C}_i^l$, is mapped to a node at the higher level graph, also called its parent node, $v_i^{l-1} := Pa(\mathcal{C}_i^l)$ and each bipartite at level $l$ is represented by an edge in the higher level, also called its parent edge, $e_i^{l-1} = Pa(\mathcal{B}_{ij}^l) = (v_i^{l-1}, v_j^{l-1})$. The weights of the self edges and the weights of the cross-edges in the parent level are determined by the sum of the weights of the edges within their corresponding community and bipartite, respectively. Therefore, the edges in the induced graphs at the higher levels have integer-valued weights: $w_{ii}^{l-1} = \sum_{e \in \mathcal{E}(\mathcal{C}_i^l)} w_e$ and $w_{ij}^{l-1} = \sum_{e \in \mathcal{E}(\mathcal{B}_{ij}^l)} w_e$, moreover sum of all edge weights remains constant in all levels so $w_0 := \sum_{e \in \mathcal{E}(\mathcal{G}^l)} w_e = |\mathcal{E}|, \forall l \in [0, ..., L]$.

This clustering process continues recursively in a bottom-up approach until a single node graph $\mathcal{G}^0$ is obtained, producing a *hierarchical graph*, defined by the set of graphs in all levels of abstractions, $\mathcal{HG} := \{\mathcal{G}^0, ...., \mathcal{G}^{L-1}, \mathcal{G}^L\}$. This forms a dendrogram tree with $\mathcal{G}^0$ being the root and $\mathcal{G}^L$ being the final graph that is generated at the leaf level. An $\mathcal{HG}$ is visualized in Figure 1a. The hierarchical tree structure enables modeling of both local and long-range interactions among nodes, as well as control over the flow of information between them, across multiple levels of abstraction. This is a key aspect of our proposed generative model. Please refer to appendix A for a list of notation definitions.

# 3 HIERARCHICAL GRAPH GENERATION

In graph generative networks, the objective is to learn a generative model, $p(\mathcal{G})$ given a set of training graphs. This work aims to establish a hierarchical multi-resolution framework for generating graphs in a coarse-to-fine fashion. In this framework, we assume that the graphs do not have node attributes, so the generative model only needs to characterize the graph topology. Given a particular node ordering $\pi$, and a hierarchical graph $\mathcal{HG} := \{\mathcal{G}^0, ...., \mathcal{G}^{L-1}, \mathcal{G}^L\}$, produced by recursively applying a graph partitioning function, $\mathcal{F}$, we can factorize the generative model using the chain rule of probability as:

$$p(\mathcal{G} = \mathcal{G}^L, \pi) = p(\{\mathcal{G}^L, \mathcal{G}^{L-1}, ..., \mathcal{G}^0\}, \pi) = p(\mathcal{G}^L, \pi \mid \{\mathcal{G}^{L-1}, ..., \mathcal{G}^0\}) ... p(\mathcal{G}^1, \pi \mid \mathcal{G}^0) \, p(\mathcal{G}^0)$$

$$= \prod_{l=0}^{L} p(\mathcal{G}^l, \pi \mid \mathcal{G}^{l-1}) \times p(\mathcal{G}^0) \tag{1}$$

In other words, the generative process involves specifying the probability of the graph at each level conditioned on its parent level graph in the hierarchy. This process is iterated recursively until the leaf level is reached. Here, the distribution of the root $p(\mathcal{G}^0) = p(\mathrm{w}^0 = w_0)$ can be simply estimated using the empirical distribution of the number of edges, $w_0 = |\mathcal{E}|$, of graphs in the training set.

Based on the partitioned structure within each level of $\mathcal{HG}$, the conditional generative probability $p(\mathcal{G}^l \mid \mathcal{G}^{l-1})$ can be decomposed into the probability of its communities and bipartite graphs as:

$$p(\mathcal{G}^l \mid \mathcal{G}^{l-1}) = p(\{\mathcal{C}_i^l \ \forall i \in \mathcal{V}(\mathcal{G}^{l-1})\} \ \cup \ \{\mathcal{B}_{ij}^l \ \forall (i,j) \in \mathcal{E}(\mathcal{G}^{l-1})\} \mid \mathcal{G}^{l-1})$$

$$\approx \prod_{i \, \in \, \mathcal{V}(\mathcal{G}^{l-1})} p(\mathcal{C}_i^l \mid \mathcal{G}^{l-1}) \times \prod_{(i,j) \in \, \mathcal{E}(\mathcal{G}^{l-1})} p(\mathcal{B}_{ij}^l \mid \mathcal{G}^{l-1}, \{\mathcal{C}_k^l\}_{\mathcal{C}_k^l \in \mathcal{G}^l}) \tag{2}$$

This decomposition is based on the assumption that, given the parent graph $\mathcal{G}^{l-1}$, generative probabilities of communities, $\mathcal{C}_i^l$, are mutually independent. Subsequent to the generation of community graphs, it further assumes that the generation probability of each bipartite can be modeled independent of the other bipartites. [1] The following theorem leverages the properties of multinomial distribution to prove the conditional independence of the components for the integer-value weighted hierarchical graph (r.t. appendix B.1 for the proof).

**Theorem 3.1.** *Let the random vector* $\mathbf{w} := [w_e]_{e \, \in \, \mathcal{E}(\mathcal{G}^l)}$ *denote the set of weights of all edges of* $\mathcal{G}^l$ *such that their sum is* $w_0 = \mathbf{1}^T \mathbf{w}$. *The joint probability of* $\mathbf{w}$ *can be described by a multinomial distribution:* $\mathbf{w} \sim Mu(\mathbf{w} \mid w_0, \boldsymbol{\theta}^l)$. *By observing that the sum of edge weights within each community* $\mathcal{C}_i^l$ *and bipartite graph* $\mathcal{B}_{ij}^l$ *are determined by the weights of their parent edges in the higher level,* $w_{ii}^{l-1}$ *and* $w_{ij}^{l-1}$ *respectively, we can establish that these components are conditionally independent and each of them follow a multinomial distribution:*

$$p(\mathcal{G}^l \mid \mathcal{G}^{l-1}) \sim \prod_{i \, \in \, \mathcal{V}(\mathcal{G}^{l-1})} Mu([w_e]_{e \, \in \, \mathcal{C}_i^l} \mid w_{ii}^{l-1}, \boldsymbol{\theta}_{ii}^l) \prod_{(i,j) \in \, \mathcal{E}(\mathcal{G}^{l-1})} Mu([w_e]_{e \, \in \, \mathcal{B}_{ij}^l} \mid w_{ij}^{l-1}, \boldsymbol{\theta}_{ij}^l) \tag{3}$$

*where* $\{\boldsymbol{\theta}_{ij}^l[e] \in [0,1], \ s.t. \ \mathbf{1}^T \boldsymbol{\theta}_{ij}^l = 1 \mid \forall \, (i,j) \in \, \mathcal{E}(\mathcal{G}^{l-1})\}$ *are the multinomial model's parameters.*

Therefore, given the parent graph at a higher level, the generation of graph at its subsequent level can be reduced to generation of its partition and bipartite sub-graphs. As illustrated in figure 1, this decomposition enables parallel generation of the communities in each level which is followed by predicting bipartite sub-graphs in that level. Each of these sub-graphs corresponds to a block in the adjacency matrix, as visualized in figure 2a, so the proposed hierarchical model generates adjacency matrix in a blocks-wise fashion and constructs the final graph topology.

## 3.1 COMMUNITY GENERATION

Based on the equation (3), the edge weights within each community can be jointly modeled using a multinomial distribution. In this work, our objective is to specify the generative probability of

---

[1]Indeed, this assumption implies that the cross dependency between communities are primarily encoded by their parent abstract graph which is reasonable where the nodes' dependencies are mostly local and are within community rather than being global.

communities in level $l$, $p(\mathcal{C}_i^l \mid \mathcal{G}^{l-1})$, as an autoregressive process, hence, we need to factorize the joint multinomial distribution of edges as a sequence of conditional distributions. Toward this goal, we show in appendix: B.2 that this multinomial distribution can be decomposed into a sequence of binomial distribution of each edge using a *stick-breaking* process. This result enables us to model the community generation as an edge-by-edge autoregressive process with $\mathcal{O}(|\mathcal{V}_\mathcal{C}|^2)$ generation steps, similar to existing algorithms such as GraphRNN (You et al., 2018) or DeepGMG (Li et al., 2018).

However, inspired by GRAN (Liao et al., 2019), a community can be generated more efficiently by generating one node at a time, reducing the sampling process to $\mathcal{O}(|\mathcal{V}_\mathcal{C}|)$ steps. This requires decomposing the generative probability of edges in a group-wise form, where the candidate edges between the $t$-th node, the new node at the $t$-th step, and the already generated graph are grouped together. In other words, this model completes the lower triangle adjacency matrix one row at a time conditioned on the already generated sub-graph and the parent-level graph. The following theorem formally derives this decomposition for multinomial distributions.

**Theorem 3.2.** *For a random counting vector* $\mathbf{w} \in \mathbb{Z}_+^E$ *with a multinomial distribution* $Mu(\mathbf{w} \mid w, \boldsymbol{\theta})$, *let's split it into $T$ disjoint groups* $\mathbf{w} = [\mathbf{u}_1, ..., \mathbf{u}_T]$ *where* $\mathbf{u}_t \in \mathbb{Z}_+^{E_t}$, $\sum_{t=1}^T E_t = E$, *and also split the probability vector accordingly as* $\boldsymbol{\theta} = [\boldsymbol{\theta}_1, ..., \boldsymbol{\theta}_T]$. *Additionally, let's define sum of all variables in the $t$-th group ($t$-th step) by a random count variable* $\mathrm{v}_t := \sum_{e=1}^{E_t} \mathrm{u}_{t,e}$. *Then, the multinomial distribution can be factorized as a chain of binomial and multinomial distributions:*

$$Mu(\mathbf{w} = [\mathbf{u}_1, ..., \mathbf{u}_T] \mid w, \boldsymbol{\theta} = [\boldsymbol{\theta}_1, ..., \boldsymbol{\theta}_T]) = \prod_{t=1}^T Bi(\mathrm{v}_t \mid \mathrm{r}_t, \eta_{\mathrm{v}_t}) \, Mu(\mathbf{u}_t \mid \mathrm{v}_t, \boldsymbol{\lambda}_t), \quad (4)$$

$$\text{where: } \mathrm{r}_t = w - \sum_{i<t} \mathrm{v}_i, \ \eta_{\mathrm{v}_t} = \frac{\mathbf{1}^\mathsf{T} \boldsymbol{\theta}_t}{1 - \sum_{i<t} \mathbf{1}^\mathsf{T} \boldsymbol{\theta}_i}, \ \boldsymbol{\lambda}_t = \frac{\boldsymbol{\theta}_t}{\mathbf{1}^\mathsf{T} \boldsymbol{\theta}_t}.$$

*Here,* $\mathrm{r}_t$ *denotes the remaining weight at $t$-th step, and the probability of binomial,* $\eta_{\mathrm{v}_t}$, *is the fraction of the remaining probability mass that is allocated to* $\mathrm{v}_t$, *i.e. the sum of all weights in the $t$-th group. The vector parameter* $\boldsymbol{\lambda}_t$ *is the normalized multinomial probabilities of all count variables in the $t$-th group. Intuitively, this decomposition of multinomial distribution can be viewed as a recursive stick-breaking process where at each step $t$: first a binomial distribution is used to determine how much probability mass to allocate to the current group, and a multinomial distribution is used to distribute that probability mass among the variables in the group. The resulting distribution is equivalent to the original multinomial distribution.* **Proof:** *Refer to appendix B.3.*

Let $\hat{\mathcal{C}}_{i,t}^l$ denote an already generated sub-graph at the $t$-th step, augmented with the set of candidate edges from the new node, $v_t(\mathcal{C}_i^l)$, to its preceding nodes, denoted by $\hat{\mathcal{E}}_t(\hat{\mathcal{C}}_{i,t}^l) := \{(t, j) \mid j < t\}$. We collect the weights of the candidate edges in the random vector $\mathbf{u}_t := [w_e]_{e \in \hat{\mathcal{E}}_t(\hat{\mathcal{C}}_{i,t}^l)}$ (that corresponds to the $t$-th row of the lower triangle of adjacency matrix $\hat{\mathbf{A}}_i^l$), where the sum of the candidate edge weights is $\mathrm{v}_t$, remaining edges' weight is $\mathrm{r}_t = w_{ii}^{l-1} - \sum_{i<t} \mathrm{v}_i$ and total edges' weight of community $\mathcal{C}_i^l$ is $w_{ii}^{l-1}$. Based on theorem 3.2, the probability of $\mathbf{u}_t$ can be characterized by the product of a binomial and a multinomial distribution. So, we need to model the parameters of the these distributions. This process is illustrated in figure 2b and figure 2 in appendix. We further increase the expressiveness of the generative network by extending this probability to a mixture model with $K$ mixtures:

$$p(\mathbf{u}_t) = \sum_{k=1}^K \boldsymbol{\beta}_k^l Bi(\mathrm{v}_t \mid \mathrm{r}_t, \eta_{t,k}^l) Mu(\mathbf{u}_t \mid \mathrm{v}_t, \boldsymbol{\lambda}_{t,k}^l) \quad (5)$$

$$\boldsymbol{\lambda}_{t,k}^l = \mathrm{softmax}\left( \mathrm{MLP}_{\boldsymbol{\theta}}^l\left( \left[ \Delta \boldsymbol{h}_{\hat{\mathcal{E}}_t(\hat{\mathcal{C}}_{i,t}^l)} \mid\mid \mathrm{pool}(\boldsymbol{h}_{\hat{\mathcal{C}}_{i,t}^l}) \mid\mid h_{Pa(\mathcal{C}_i^l)} \right] \right) \right)[k, :] \quad (6)$$

$$\eta_{t,k}^l = \mathrm{sigmoid}\left( \mathrm{MLP}_\eta^l\left( \left[ \mathrm{pool}(\boldsymbol{h}_{\hat{\mathcal{C}}_{i,t}^l}) \mid\mid h_{Pa(\mathcal{C}_i^l)} \right] \right) \right)[k]$$

$$\boldsymbol{\beta}^l = \mathrm{softmax}\left( \mathrm{MLP}_\beta^l\left( \left[ \mathrm{pool}(\boldsymbol{h}_{\hat{\mathcal{C}}_{i,t}^l}) \mid\mid h_{Pa(\mathcal{C}_i^l)} \right] \right) \right)$$

Where $\Delta \boldsymbol{h}_{\hat{\mathcal{E}}_t(\hat{\mathcal{C}}_{i,t}^l)}$ is a $|\hat{\mathcal{E}}_t(\hat{\mathcal{C}}_{i,t}^l)| \times d_h$ dimensional matrix, consisting of the set of candidate edge embeddings $\{\Delta h_{(t,s)} := h_t - h_s \mid \forall (t, s) \in \hat{\mathcal{E}}_t(\hat{\mathcal{C}}_{i,t}^l)\}$, $\boldsymbol{h}_{\hat{\mathcal{C}}_{i,t}^l}$ is a $t \times d_h$ matrix of node embeddings

of $\hat{\mathcal{C}}_{i,t}^l$ learned by a GNN model: $\boldsymbol{h}_{\hat{\mathcal{C}}_{i,t}^l} = \mathrm{GNN}_{com}^l(\hat{\mathcal{C}}_{i,t}^l)$. Here, the graph level representation is obtained by the $\mathrm{addpool}()$ aggregation function and the mixture weights are denoted by $\boldsymbol{\beta}^l$. In order to produce $K \times |\mathcal{E}_t(\mathcal{C}_i^l)|$ dimensional matrix of multinomial probabilities, the $\mathrm{MLP}_{\boldsymbol{\theta}}^l()$ network acts at the edge level, while $\mathrm{MLP}_{\eta_v}^l()$ and $\mathrm{MLP}_{\beta}^l()$ act at the graph level to produce the binomial probabilities and $K$ dimensional arrays for $K$ mixture models, respectively. All of these MLP networks are built by two hidden layers with $\mathrm{ReLU}()$ activation functions.

In the generative model of each community $\mathcal{C}_i^l$, the embedding of its parent node, $h_{Pa(\mathcal{C}_i^l)}$, is used as the context, and is concatenated to the node and edge embeddings at level $l$. The operation $\begin{bmatrix} \boldsymbol{x} \, || \, y \end{bmatrix}$ concatenates vector $y$ to each row of matrix $\boldsymbol{x}$. Since parent level reflects global structure of the graph, concatenating its features enriches the node and edge embeddings by capturing long-range interactions and global structure of the graph, which is important for generating local components.

## 3.2   Bipartite Generation

Once all the communities in level $l$ are generated, the edges of all bipartite graphs at that level can be predicted simultaneously. An augmented graph $\hat{\mathcal{G}}^l$ composed of all the communities, $\{\mathcal{C}_i^l \;\; \forall i \in \mathcal{V}(\mathcal{G}^{l-1})\}$, and the candidate edges of all bipartites, $\{\mathcal{B}_{ij}^l \;\; \forall (i,j) \in \mathcal{E}(\mathcal{G}^{l-1})\}$. Node and edge embeddings are encoded by $\mathrm{GNN}_{bp}^l(\hat{\mathcal{G}}^l)$. We similarly extend the multinomial distribution of a bipartite, in (11), using a mixture model to express its generative probability:

$$p(\mathbf{w} := \hat{\mathcal{E}}(\mathcal{B}_{ij}^l)) = \sum_{k=1}^K \boldsymbol{\beta}_k^l \mathrm{Mu}(\mathbf{w} \mid w_{ij}^{l-1}, \boldsymbol{\theta}_{ij,k}^l)$$

$$\boldsymbol{\theta}_{ij,k}^l = \mathrm{softmax}\left(\mathrm{MLP}_{\boldsymbol{\theta}}^l\left(\left[\Delta\boldsymbol{h}_{\hat{\mathcal{E}}(\mathcal{B}_{ij}^l)} \, || \, \Delta h_{Pa(\mathcal{B}_{ij}^l)}\right]\right)\right)[k,:] \qquad (7)$$

$$\boldsymbol{\beta}^l = \mathrm{softmax}\left(\mathrm{MLP}_{\beta}^l\left(\left[\mathrm{pool}(\Delta\boldsymbol{h}_{\hat{\mathcal{E}}(\mathcal{B}_{ij}^l)}) \, || \, \Delta h_{Pa(\mathcal{B}_{ij}^l)}\right]\right)\right)$$

where the random vector $\mathbf{w} := [w_e]_{e \, \in \, \hat{\mathcal{E}}(\mathcal{B}_{ij}^l)}$ is the set of weights of all candidate edges in bipartite $\mathcal{B}_{ij}^l$ , and $\Delta\boldsymbol{h}_{Pa(\mathcal{B}_{ij}^l)}$ are the parent edge embeddings of the bipartite graph. By parametrizing the distribution of bipartite graphs based on both the generated communities and the parent graph, HiGen can effectively capture the interdependence between bipartites and communities.

**Node Feature Encoding:**   To encode node embeddings, we extend GraphGPS proposed by Rampášek et al. (2022). GraphGPS combines local message-passing with global attention mechanism and uses positional and structural encoding for nodes and edges to construct a more expressive and a scalable graph transformer (GT) (Dwivedi & Bresson, 2020). We employ GraphGPS as the GNN of the parent graph, $\mathrm{GNN}^{l-1}(\mathcal{G}^{l-1})$. However, to apply GraphGPS on augmented graphs of bipartites, $\mathrm{GNN}_{bp}^l(\hat{\mathcal{G}}^l)$, we use distinct initial edge features to distinguish augmented (candidate) edges from real edges. Furthermore, for bipartite generation, the attention scores in the Transformers of the augmented graph $\hat{\mathcal{G}}^l$ are masked to restrict attention only to connected communities. Moreover, for the community generation, we employ the GNN with attentive messages model, proposed in (Liao et al., 2019), as $\mathrm{GNN}_{com}^l$. The details of model architecture are provided in appendix C.1.

## 4   Related Work

In order to deal with hierarchical structures in molecular graphs, a generative process was proposed by Jin et al. (2020) which recursively selects motifs, the basic building blocks, from a set and predicts their attachment to the emerging molecule. However, this method requires prior domain-specific knowledge and relies on molecule-specific graph motifs. Additionally, the graphs are only abstracted into two levels, and component generation cannot be performed in parallel. In (Kuznetsov & Polykovskiy, 2021), a hierarchical normalizing flow model for molecular graphs was introduced, where new molecules are generated from a single node by recursively dividing each node into two. However, the merging and splitting of pairs of nodes in this model is based on the node's neighborhood without accounting for the diverse community structure of graphs, hence the graph generation of this model is inherently limited. Shirzad et al. (2022) proposed a graph generation framework based

on tree decomposition that reduces upper bound on the maximum number of sampling decisions. However, this model is limited to a single level of abstraction with tree structure and requires $\mathcal{O}(nk)$ generation steps where $k$ represents the width of the tree decomposition, hence its scalability is limited to medium-sized graphs. In contrast, HiGen's ability to employ different generation models for community and inter-community subgraphs at multiple levels of abstraction is a key advantage that enhances its expressiveness.

## 5 EXPERIMENTS

In our empirical studies, we compare the proposed hierarchical graph generative network against state-of-the-art autoregressive models: GRAN and GraphRNN models, diffusion models: DiGress (Vignac et al., 2022), GDSS (Jo et al., 2022), GraphARM (Kong et al., 2023) and EDGE (Chen et al., 2023) , and a GAN-based model: SPECTRE (Martinkus et al., 2022), on a range of synthetics and real datasets of various sizes.

**Datasets:** We used 5 different benchmark graph datasets: (1) the synthetic *Stochastic Block Model (SBM)* dataset consisting of 200 graphs with 2-5 communities each with 20-40 nodes (Martinkus et al., 2022); (2) the *Protein* including 918 protein graphs, each has 100 to 500 nodes representing amino acids that are linked if they are closer than 6 Angstroms (Dobson & Doig, 2003), (3) the *Enzyme* that has 587 protein graphs of 10-125 nodes, representing protein tertiary structures of the enzymes from the BRENDA database (Schomburg et al., 2004) and (4) the *Ego* dataset containing 757 3-hop ego networks with 50-300 nodes extracted from the CiteSeer dataset (Sen et al., 2008). (5) *Point Cloud* dataset, which consists of 41 $3D$ point clouds of household objects. The dataset consists of large graphs with up to 5k nodes and approximately 1.4k nodes on average (Neumann et al., 2013).

**Graph Partitioning** Different algorithms approach the problem of graph partitioning (clustering) using various clustering quality functions. Two commonly used families of such metrics are modularity and cut-based metrics (Tsitsulin et al., 2020). Although optimizing modularity metric is an NP-hard problem, it is well-studied in the literature and several graph partitioning algorithm based on this metric have been proposed. For example, the Louvain algorithm (Blondel et al., 2008) starts with each node as its community and then repeatedly merges communities based on the highest increase in modularity until no further improvement can be made. This heuristic algorithm is computationally efficient and scalable to large graphs for community detection. Moreover, a spectral relaxation of modularity metrics has been proposed in Newman (2006a;b) which results in an analytical solution for graph partitioning. Additionally, an unsupervised GNN-based pooling method inspired by this spectral relaxation was proposed for partitioning graphs with node attributes (Tsitsulin et al., 2020). As the modularity metric is based on the graph structure, it is well-suited for our problem. Therefore, we employed the Louvain algorithm to get a hierarchical clustering of the graph nodes in the datasets and then spliced out the intermediate levels to achieve $\mathcal{HG}$s with uniform depth of $L = 2$.

**Model Architecture** In our experiments, the GNN models consist of 8 layers of GraphGPS layers, and the input node features are augmented with positional and structural encoding. This encoding includes the first 8 eigenvectors related to the smallest non-zero eigenvalues of the Laplacian and the diagonal of the random-walk matrix up to 8 steps. Each hierarchical level employs its own GNN and output models. For more architectural details, please refer to Appendix C.1 and D.

We conducted experiments using the proposed hierarchical graph generative network (HiGen) model with two variants for the output distribution of the leaf edges: 1) **HiGen**: the probability of the community edges' weights at the leaf level are modeled by mixture of Bernoulli, using $\mathrm{sigmoid}()$ activation in (6), since the leaf levels in our experiments have binary edges weights. 2)**HiGen-m**: the model uses a mixture of multinomial distributions (5) to describe the output distribution for all levels. In this case, we observed that modeling the probability parameters of edge weights at the leaf level, denoted by $\boldsymbol{\lambda}_{t,k}^L$ in (6), by a multi-hot activation function, defined as $\sigma(\mathbf{z})_i := \mathrm{sigmoid}(z_i)/\sum_{j=1}^{K} \mathrm{sigmoid}(z_j)$ where $\sigma : \mathbb{R}^K \to (K-1)$-simplex, provided slightly better performance than the standard $\mathrm{softmax}()$ function. However, for both HiGen and HiGen-m, the probabilities of the integer-valued edges at the higher levels are still modeled by the standard $\mathrm{softmax}()$ function.[2]

---

[2]As the leaf levels have binary edge weights while the sum of their weights is determined by their parent edge, a possible extension to this work could be using the cardinality potential model (Hajimirsadeghi et al.,

Table 1: Comparison of generation metrics on benchmark datasets. The baseline results for SBM and Protein graphs are obtained from (Martinkus et al., 2022; Vignac et al., 2022), the results for enzyme graphs are obtained from (Kong et al., 2023). and the scores for Ego are from (Chen et al., 2023). For Ego we report GNN-based performance metrics: GNN RBF and Frechet Distance (FD) besides structure-based statistics. For all the scores, the smaller the better. Best results are indicated in bold and the second best methods are underlined. "-": not applicable due to resource issue or not reported in the reference papers. On the right side, the samples from HiGen are depicted where the communities are distinguished with different colors at 2 levels.

| Model | Stochastic block model | | | | Protein | | | |
|---|---|---|---|---|---|---|---|---|
| | Deg. | Clus. | Orbit | Spec. | Deg. | Clus. | Orbit | Spec. |
| **GraphRNN** | 0.0055 | 0.0584 | 0.0785 | 0.0065 | 0.0040 | 0.1475 | 0.5851 | 0.0152 |
| **GRAN** | 0.0113 | 0.0553 | 0.0540 | 0.0054 | 0.0479 | 0.1234 | 0.3458 | 0.0125 |
| **SPECTRE** | 0.0015 | 0.0521 | 0.0412 | 0.0056 | 0.0056 | 0.0843 | 0.0267 | 0.0052 |
| **DiGress** | **0.0013** | 0.0498 | 0.0433 | - | - | - | - | - |
| **HiGen-m** | 0.0017 | 0.0503 | 0.0604 | 0.0068 | 0.0041 | 0.109 | 0.0472 | 0.0061 |
| **HiGen** | 0.0019 | **0.0498** | **0.0352** | **0.0046** | **0.0012** | **0.0435** | **0.0234** | **0.0025** |

| Model | Enzyme | | | Model | Ego | | | | |
|---|---|---|---|---|---|---|---|---|---|
| | Deg. | Clus. | Orbit | | Deg. | Clus. | Orbit | GNN RBF | FD |
| **GraphRNN** | 0.017 | 0.062 | 0.046 | **GraphRNN** | 0.0768 | 1.1456 | 0.1087 | 0.6827 | 90.57 |
| **GRAN** | 0.054 | 0.087 | 0.033 | **GRAN** | 0.5778 | 0.3360 | 0.0406 | 0.2633 | 489.96 |
| **GDSS** | 0.026 | 0.061 | 0.009 | **GDSS** | 0.8189 | 0.6032 | 0.3315 | 0.4331 | 60.61 |
| **SPECTRE** | 0.136 | 0.195 | 0.125 | **DiscDDPM** | 0.4613 | 0.1681 | 0.0633 | 0.1561 | 42.80 |
| **DiGress** | **0.004** | 0.083 | 0.002 | **DiGress** | 0.0708 | 0.0092 | 0.1205 | 0.0489 | 18.68 |
| **GraphARM** | 0.029 | 0.054 | 0.015 | **EDGE** | 0.0579 | 0.1773 | 0.0519 | 0.0658 | 15.76 |
| **HiGen-m** | 0.027 | 0.157 | 1.2e-3 | **HiGen-m** | 0.114 | 0.0378 | 0.0535 | **0.0420** | 12.2 |
| **HiGen** | 0.012 | **0.038** | **7.2e-4** | **HiGen** | **0.0472** | **0.0031** | **0.0387** | 0.0454 | **5.24** |

SBM          Protein

Enzyme       Ego

**Metrics** To evaluate the graph generative models, we compare the distributions of four different graph structure-based statistics between the ground truth and generated graphs: (1) degree distributions, (2) clustering coefficient distributions, (3) the number of occurrences of all orbits with four nodes, and (4) the spectra of the graphs by computing the eigenvalues of the normalized graph Laplacian. The first three metrics capture local graph statistics, while the spectra represents global structure. The maximum mean discrepancy (MMD) score over these statistics are used as the metrics. While Liu et al. (2019) computed MMD scores using the computationally expensive Gaussian earth mover's distance (EMD) kernel, Liao et al. (2019) proposed using the total variation (TV) distance as an alternative measure. TV distance is much faster and still consistent with the Gaussian EMD kernel. Recently, O'Bray et al. (2021) suggested using other efficient kernels such as an RBF kernel, or a Laplacian kernel, or a linear kernel. Moreover, Thompson et al. (2022) proposed new evaluation metrics for comparing graph sets using a random-GNN approach where GNNs are employed to extract meaningful graph features. However, in this work, we follow the experimental setup and evaluation metrics of (Liao et al., 2019), except for the enzyme dataset where we use a Gaussian EMD kernel to be consistent with the results reported in (Jo et al., 2022). GNN-based performance metrics of HiGen model are also reported in appendix E.

The performance metrics of the proposed HiGen models are reported in Table 1, together with generated graph samples of HiGen. The results demonstrate that HiGen effectively captures graph statistics and achieves state-of-the-art on all the benchmarks graphs across various generation metrics. This improvement in both local and global properties of the generated graphs highlights the effectiveness of the hierarchical graph generation approach, which models communities and cross-community interactions separately. A comparison of sampling times for the model can be found in Appendix D.2. Additionally, Appendix E contains visual comparisons of generated graphs by the HiGen models and an experimental evaluation of various node ordering and partitioning functions.

For *point cloud dataset*, the augmented graph of bipartites, $\hat{\mathcal{G}}^l$, contains very large number of candidate edges, leading to out-of-memory during training. To overcome this challenge, we adopted a sub-graph sampling strategy, allowing us to generate one or a subset of bipartites at a time to ensure memory constraints were met. In our experiments, we sequenced the generation of bipartites based on the index of their parent edges in the parent graph.

---

2015), which is derived to model the distribution over the set of binary random variables, to model the edge weight at the leaf level.

In this context, when generating the edges of $\mathcal{B}_{ij}^l$, the augmented graph encompassed all preceding communities $(\mathcal{C}_k^l \;\; \forall k \leq \max(i,j))$, bipartites $(\mathcal{B}_{xy}^l \;\; \forall (x,y) \leq (i,j))$ and the candidate edges of $\mathcal{B}_{ij}^l$. We trained different models for hierarchical depth of $L=2$ and $L=3$. The results of this approach, referred to as HiGen-s, in Table 11 highlights that HiGen-s outperforms the baselines while other baseline models are not applicable due to out-of-memory and computational complexity.

Table 2: Comparison of generation metrics on benchmark 3D point cloud. The baseline results are from (Liao et al., 2019)

| Model | 3D Point Cloud | | | |
| | Deg. ↓ | Clus. ↓ | Orbit↓ | Spec. ↓ |
| --- | --- | --- | --- | --- |
| **Erdos-Renyi** | 3.1e-01 | 1.22 | 1.27 | 4.26e-02 |
| **GRAN** | **1.75e-02** | 5.1e-01 | 2.1e-01 | 7.45e-03 |
| **HiGen-s (L=2)** | 3.48e-02 | **2.82e-01** | 3.45e-02 | 5.46e-03 |
| **HiGen-s (L=3)** | 4.97e-02 | 3.19e-01 | **1.97e-02** | **5.2e-03** |

This modification can also address a potential limitation related to edge independence when generating all the inter-communities simultaneously. However, it's important to note that the significance of edge independence is more prominent in high-density graphs like community generations, (Chanpuriya et al., 2021), whereas its impact is less significant in sparser inter-communities of hierarchical approach. This is evident by the performance improvement observed in our experiments.

## 6 Discussion and Conclusion

**Node ordering sensitivity:**  The predefined ordering of dimensions can be crucial for training autoregressive (AR) models (Vinyals et al., 2015), and this sensitivity to node orderings is particularly pronounced in autoregressive graph generative models (Liao et al., 2019; Chen et al., 2021). However, in the proposed approach, the graph generation process is divided into the generation of multiple small partitions, performed sequentially across the levels, rather than generating the entire graph by a single AR model. Therefore, given an ordering for the parent level, the graph generation depends only on the permutation of the nodes within the graph communities rather than the node ordering of the entire graph. In other words, the proposed method is invariant to a large portion of possible node permutations, and therefore the set of distinctive adjacency matrices is much smaller in HiGen. For example, the node ordering $\pi_1 = [v_1, v_2, v_3, v_4]$ with clusters $\mathcal{V}_{\mathcal{G}_1} = \{v_1, v_2\}$ and $\mathcal{V}_{\mathcal{G}_2} = \{v_3, v_4\}$ has a similar hierarchical graph as $\pi_2 = [v_1, v_3, v_2, v_4]$, since the node ordering within the communities is preserved at all levels. Formally, let $\{\mathcal{C}_i^l \;\; \forall i \in \mathcal{V}_{\mathcal{G}^{l-1}}\}$ be the set of communities at level $l$ produced by a deterministic partitioning function, where $n_i^l = |\mathcal{V}(\mathcal{C}_i^l)|$ denotes the size of each partition. The upper bound on the number of distinct node orderings in an HG generated by the proposed process is then reduced to $\prod_{l=1}^{L} \prod_i n_i^l!$. [3]

The proposed hierarchical model allows for highly parallelizable training and generation. Specifically, let $n_c := \max_i(|\mathcal{C}_i|)$ denote the size of the largest community, then, it only requires $\mathcal{O}(n_c \log n)$ sequential steps to generate a graph of size $n$.

**Block-wise generation:**  GRAN generates graphs one block of nodes at a time using an autoregressive approach, but its performance declines with larger block sizes. This happens because adjacent nodes in an ordering might not be related and could belong to different structural clusters. In contrast, our method generates node blocks within communities with strong connections and predicts cross-links between communities using a separate model. This allows our approach to capture both local relationships within a community and global relationships across communities, enhancing the expressiveness of the graph generative model.

**Conclusion**  This work introduces a novel graph generative model, HiGen, which effectively captures the multi-scale and community structures inherent in complex graphs. By leveraging a hierarchical approach that focuses on community-level generation and cross-community link prediction, HiGen demonstrates significant improvements in performance and scalability compared to existing models, bridging the gap between one-shot and autoregressive graph generative models. Experimental results on benchmark datasets demonstrate that HiGen achieves state-of-the-art performance in terms of graph generation across various metrics. The hierarchical and block-wise generation strategy of HiGen enables scaling up graph generative models to large and complex graphs, making it adaptable to emerging generative paradigms.

---

[3]It is worth noting that all node permutations do not result in distinctive adjacency matrices due to the automorphism property of graphs (Liao et al., 2019; Chen et al., 2021). Therefore, the number of node permutations provides an upper bound rather than an exact count.

REPRODUCIBILITY

To ensure reproducibility, we provide comprehensive documentation of key elements in this work. The complete proofs of the theorems, community generative model, bipartite distribution, GNN architectures, and the loss function can be found in Appendices A, B, and C. Furthermore, Section 5 and Appendix D include experimental details, model architectures, benchmark graph datasets, their statistics, the computational resource and the graph partitioning function. These resources collectively facilitate the reproducibility of our research findings.

ACKNOWLEDGMENTS

We would like to thank Fatemeh Fani Sani for preparing the schematic figures.

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

# A  NOTATION DEFINITION

| Notations | Brief definition and interpretation |
|---|---|
| $\mathcal{G} = (\mathcal{V}, \mathcal{E})$ | A graph with nodes (vertices) $\mathcal{V}$ and edges $\mathcal{E}$ |
| $\mathcal{F} : \mathcal{V} \to \{1, ..., c\}$ | a graph partitioning function that partitions a graph into $c$ communities (a.k.a. cluster or modules) |
| $\mathcal{C}_i^l = \left(\mathcal{V}(\mathcal{C}_i^l), \mathcal{E}(\mathcal{C}_i^l)\right)$ | $i$-th community graph (a cluster of nodes) at level $l$ |
| $\mathbf{A}^l, \mathbf{A}_i^l, \mathbf{A}_{ij}^l$ | adjacency matrix of a graph, community and bipartite |
| $\mathcal{B}_{ij}^l = \left(\mathcal{V}(\mathcal{C}_i^l), \mathcal{V}(\mathcal{C}_j^l), \mathcal{E}(\mathcal{B}_{ij}^l)\right)$ | a bipartite (or cross-community) graph composed of cross-links between neighboring communities |
| $v_i^{l-1} := Pa(\mathcal{C}_i^l)$ | parent node of $\mathcal{C}_i^l$ at the higher level |
| $e_i^{l-1} = Pa(\mathcal{B}_{ij}^l) = (v_i^{l-1}, v_j^{l-1})$ | parent edge of $\mathcal{B}_{ij}^l$ at the higher level |
| $w_{ii}^{l-1} = \sum_{e \in \mathcal{E}(\mathcal{C}_i^l)} w_e$ | weight of edge $(v_i^{l-1}, v_i^{l-1})$: sum of the weights of the edges within their child community |
| $w_{ij}^{l-1} = \sum_{e \in \mathcal{E}(\mathcal{B}_{ij}^l)} w_e$ | weight of edge $(v_i^{l-1}, v_j^{l-1})$: sum of the weights of the edges within their child bipartite |
| $w_0 := \sum_{e \in \mathcal{E}(\mathcal{G}^l)} w_e = |\mathcal{E}|$ | sum of all edge weights that remains constant in all levels |
| $\mathcal{HG} := \{\mathcal{G}^0, ...., \mathcal{G}^{L-1}, \mathcal{G}^L\}$ | the set of graphs in all levels, where $\mathcal{G}^0$ is a single node root graph and $\mathcal{G}^L$ is the final graph at the leaf level. |
| $\text{Mu}(\mathbf{u} \mid \text{v}, \boldsymbol{\lambda})$ | Multinomial distribution of random vector $\mathbf{u}$, with parameters $(\text{v}, \boldsymbol{\lambda})$ |
| $\text{Bi}(\text{v} \mid \text{r}, \eta)$ | Binomial distribution of random variable v, with parameters $(\text{r}, \eta)$ |
| $\hat{\mathcal{E}}_t(\hat{\mathcal{C}}_{i,t}^l) = \{(t, j) \mid j < t\}$ | set of candidate (augmented) edges from the new node, $v_t(\mathcal{C}_i^l)$, to its preceding nodes of community |
| $\hat{\mathcal{C}}_{i,t}^l = \left(\mathcal{V}(\mathcal{C}_{i,t-1}^l) \cup v_t(\mathcal{C}_i^l), \mathcal{E}(\mathcal{C}_{i,t-1}^l) \cup \hat{\mathcal{E}}_t\right)$ | an already generated community at the $t$-th step, augmented with the set of candidate edges. Its size is $t$ |
| $\mathbf{u}_t := [w_e]_{e \in \hat{\mathcal{E}}_t(\hat{\mathcal{C}}_{i,t}^l)}$ | Random vector of weights of the candidate edges in $\hat{\mathcal{C}}_{i,t}^l$ (the $t$-th row of the lower triangle $\hat{\mathbf{A}}_i^l$) |
| $\text{v}_t = \mathbf{1}^\top \mathbf{u}_t$ | sum of the candidate edge weights at step $t$. |
| $\text{r}_t = w_{ii}^{l-1} - \sum_{i<t} \text{v}_i$ | remaining edges' weight at step $t$. |
| $\boldsymbol{h}_{\hat{\mathcal{C}}_{i,t}^l}$ | $t \times d_h$ matrix of node embeddings of $\hat{\mathcal{C}}_{i,t}^l$ |
| $\Delta \boldsymbol{h}_{\hat{\mathcal{E}}_t(\hat{\mathcal{C}}_{i,t}^l)}$ | $|\hat{\mathcal{E}}_t(\hat{\mathcal{C}}_{i,t}^l)| \times d_h$ dimensional matrix of candidate edge embeddings |
| $\hat{\mathcal{G}}^l$ | An augmented graph composed of all the communities, $\{\mathcal{C}_i^l \ \forall i \in \mathcal{V}(\mathcal{G}^{l-1})\}$, and the candidate edges of all bipartites, $\{\mathcal{B}_{ij}^l \ \forall (i,j) \in \mathcal{E}(\mathcal{G}^{l-1})\}$. It is used for bipartite generation. |

# B  PROBABILITY DISTRIBUTION OF COMMUNITIES AND BIPARTITES

**Theorem B.1.** *Given a graph $\mathcal{G}$ and an ordering $\pi$, assuming there is a deterministic function that provides the corresponding high-level graphs in a hierarchical order as $\{\mathcal{G}^L, \mathcal{G}^{L-1}, ..., \mathcal{G}^0\}$, then:*

$$p(\mathcal{G} = \mathcal{G}^L, \pi) = p(\{\mathcal{G}^L, \mathcal{G}^{L-1}, ..., \mathcal{G}^0\}, \pi) = p(\mathcal{G}^L, \pi \mid \{\mathcal{G}^{L-1}, ..., \mathcal{G}^0\}) \, ... \, p(\mathcal{G}^1, \pi \mid \mathcal{G}^0) \, p(\mathcal{G}^0)$$

$$= \prod_{l=0}^{L} p(\mathcal{G}^l, \pi \mid \mathcal{G}^{l-1}) \times p(\mathcal{G}^0) \tag{8}$$

*Proof.* The factorization is derived by applying the chain rule of probability and last equality holds as the graphs at the coarser levels are produced by a partitioning function acting on the finer level graphs. Overall, this hierarchical generative model exhibits a Markovian structure. □

### B.1 PROOF OF THEOREM 3.1

**Lemma B.2.** *Given the sum of counting variables in the groups, the groups are independent and each of them has multinomial distribution:*

$$p(\mathbf{w} = [\mathbf{u}_1, \ ..., \mathbf{u}_M] | \{\mathbf{v}_1, ..., \mathbf{v}_M\}) = \prod_{m=1}^{M} Mu(\mathbf{v}_m, \ \boldsymbol{\lambda}_m)$$

$$\text{where: } \boldsymbol{\lambda}_m = \frac{\boldsymbol{\theta}_m}{\mathbf{1}^T \ \boldsymbol{\theta}_m}$$

*Here, probability vector (parameter)* $\boldsymbol{\lambda}_m$ *is the normalized multinomial probabilities of the counting variables in the* $m$-*th group.*

*Proof.*

$$p(\mathbf{w} | \{\mathbf{v}_1, ..., \mathbf{v}_M\}) = \frac{p(\mathbf{w})}{p(\{\mathbf{v}_1, ..., \mathbf{v}_M\})} I(\mathbf{v}_1 = \mathbf{1}^T \ \mathbf{u}_1, \ ..., \mathbf{v}_M = \mathbf{1}^T \ \mathbf{u}_M)$$

$$= \frac{\frac{w!}{\prod_{i=1}^{E} \mathbf{w}_i!} \prod_{i=1}^{E} \boldsymbol{\theta}_i^{\mathbf{w}_i}}{\frac{w!}{\prod_{i=1}^{M} \mathbf{v}_i!} \prod_{i=1}^{M} \alpha_i^{\mathbf{v}_i}} I(\mathbf{v}_1 = \mathbf{1}^T \ \mathbf{u}_1, \ ..., \mathbf{v}_M = \mathbf{1}^T \ \mathbf{u}_M)$$

$$= \frac{\frac{w!}{\prod_{i=1}^{E} \mathbf{w}_i!} \boldsymbol{\theta}_1^{\mathbf{w}_1} ... \boldsymbol{\theta}_E^{\mathbf{w}_E}}{\frac{w!}{\prod_{i=1}^{M} \mathbf{v}_i!} (\mathbf{1}^T \ \boldsymbol{\theta}_1)^{\mathbf{v}_1} ... (\mathbf{1}^T \ \boldsymbol{\theta}_M)^{\mathbf{v}_M}}$$

$$= \frac{\mathbf{v}_1!}{\prod_{i=1}^{E_1} \mathbf{u}_{1,i}!} \prod_{i=1}^{E_1} \boldsymbol{\lambda}_{1,i}^{\mathbf{u}_{1,i}} \times ... \times \frac{\mathbf{v}_M!}{\prod_{i=1}^{E_M} \mathbf{u}_{M,i}!} \prod_{i=1}^{E_1} \boldsymbol{\lambda}_{M,i}^{\mathbf{u}_{M,i}}$$

$$= Mu(\mathbf{v}_1, \ \boldsymbol{\lambda}_1) \times ... \times Mu(\mathbf{v}_M, \ \boldsymbol{\lambda}_M)$$

□

In a hierarchical graph, the edges has non-negative integer valued weights while the sum of all the edges in community $\mathcal{C}_i^l$ and bipartite graph $\mathcal{B}_{ij}^l$ are determined by their corresponding edges in the parent graph, *i.e.* $w_{ii}^{l-1}$ and $w_{ij}^{l-1}$ respectively. Let the random vector $\mathbf{w} := [w_e]_{e \in \mathcal{E}(\mathcal{G}^l)}$ denote the set of weights of all edges of $\mathcal{G}^l$ such that $w_0 = \mathbf{1}^T \mathbf{w}$, its joint probability can be described as a multinomial distribution:

$$\mathbf{w} \sim Mu(\mathbf{w} \mid w_0, \boldsymbol{\theta}^l) = \frac{w_0!}{\prod_{e=1}^{|\mathcal{E}(\mathcal{G}^l)|} \mathbf{w}[e]!} \prod_{e=1}^{|\mathcal{E}(\mathcal{G}^l)|} (\boldsymbol{\theta}^l[e])^{\mathbf{w}[e]}, \tag{9}$$

where $\{\boldsymbol{\theta}^l[e] \in [0, 1], \text{ s.t. } \mathbf{1}^T \boldsymbol{\theta}^l = 1\}$ are the parameters of the multinomial distribution.[4] Therefore, based on lemma B.2 these components are conditionally independent and each of them has a multinomial distribution:

$$p(\mathcal{G}^l \mid \mathcal{G}^{l-1}) \sim \prod_{i \in \mathcal{V}(\mathcal{G}^{l-1})} Mu([w_e]_{e \in \mathcal{C}_i^l} \mid w_{ii}^{l-1}, \boldsymbol{\theta}_{ii}^l) \times \prod_{(i,j) \in \mathcal{E}(\mathcal{G}^{l-1})} Mu([w_e]_{e \in \mathcal{B}_{ij}^l} \mid w_{ij}^{l-1}, \boldsymbol{\theta}_{ij}^l)$$

where $\{\boldsymbol{\theta}_{ij}^l[e] \in [0, 1], \text{ s.t. } \mathbf{1}^T \boldsymbol{\theta}_{ij}^l = 1 \mid \forall (i, j) \in \mathcal{E}(\mathcal{G}^{l-1})\}$ are the parameters of the model.

Therefore, the log-likelihood of $\mathcal{G}^l$ can be decomposed as the log-likelihood of its sub-structures:

$$\log p_{\phi^l}(\mathcal{G}^l \mid \mathcal{G}^{l-1}) = \sum_{i \in \mathcal{V}_{\mathcal{G}^{l-1}}} \log p_{\phi^l}(\mathcal{C}_i^l \mid \mathcal{G}^{l-1}) + \sum_{(i,j) \in \mathcal{E}_{\mathcal{G}^{l-1}}} \log p_{\phi^l}(\mathcal{B}_{ij}^l \mid \mathcal{G}^{l-1}) \tag{10}$$

□

---

[4]It is analogous to the random trial of putting $n$ balls into $k$ boxes, where the joint probability of the number of balls in all the boxes follows the multinomial distribution.

**Bipartite distribution:** Let's denote the set of weights of all candidate edges of the bipartite $\mathcal{B}_{ij}^l$ by a random vector $\mathbf{w} := [w_e]_{e \in \mathcal{E}(\mathcal{B}_{ij}^l)}$, its probability can be described as

$$\mathbf{w} \sim \text{Mu}(\mathbf{w} \mid w_{ij}^{l-1}, \boldsymbol{\theta}_{ij}^l) = \frac{w_{ij}^{l-1}!}{\prod_{e=1}^{|\mathcal{E}(\mathcal{B}_{ij}^l)|} \mathbf{w}[e]!} \prod_{e=1}^{|\mathcal{E}(\mathcal{B}_{ij}^l)|} (\boldsymbol{\theta}_{ij}^l[e])^{\mathbf{w}[e]} \tag{11}$$

where $\{\boldsymbol{\theta}_{ij}^l[e] \mid \boldsymbol{\theta}_{ij}^l[e] \geq 0, \ \sum \boldsymbol{\theta}_{ij}^l[e] = 1\}$ are the parameter of the distribution, and the multinomial coefficient $\frac{n!}{\prod \mathbf{w}[e]!}$ is the number of ways to distribute the total weight $w_{ij}^{l-1} = \sum_{e=1}^{|\mathcal{E}(\mathcal{B}_{ij}^l)|} \mathbf{w}[e]$ into all candidate edges of $\mathcal{B}_{ij}^l$.

**Community distribution:** Similarly, the probability distribution of the set of candidate edges for each community can be modeled jointly by a multinomial distribution but as our objective is to model the generative probability of communities in each level as an autoregressive process we are interested to decomposed this probability distribution accordingly.

## B.2 Generating a community as an edge-by-edge autoregressive process

**Lemma B.3.** *A random counting vector $\mathbf{w} \in \mathbb{Z}_+^E$ with a multinomial distribution can be recursively decomposed into a sequence of binomial distributions as follows:*

$$Mu(\mathbf{w}_1, ..., \mathbf{w}_E \mid w, [\theta_1, ..., \theta_E]) = \prod_{e=1}^{E} Bi(\mathbf{w}_e \mid w - \sum_{i<e} \mathbf{w}_i, \hat{\theta}_e), \tag{12}$$

$$where: \hat{\theta}_e = \frac{\theta_e}{1 - \sum_{i<e} \theta_i}$$

*This decomposition is known as a stick-breaking process, where $\hat{\theta}_e$ is the fraction of the remaining probabilities we take away every time and allocate to the $e$-th component (Linderman et al., 2015).*

## B.3 Proof of Theorem 3.2

For a random counting vector $\mathbf{w} \in \mathbb{Z}_+^E$ with multinomial distribution $\text{Mu}(w, \boldsymbol{\theta})$, let's split it into $M$ disjoint groups $\mathbf{w} = [\mathbf{u}_1, ..., \mathbf{u}_M]$ where $\mathbf{u}_m \in \mathbb{Z}_+^{E_m}$, $\sum_{m=1}^{M} E_m = E$, and also split the probability vector as $\boldsymbol{\theta} = [\boldsymbol{\theta}_1, ..., \boldsymbol{\theta}_M]$. Additionally, let's define sum of all weights in $m$-th group by a random variable $\mathbf{v}_m := \sum_{e=1}^{E_m} \mathbf{u}_{m,e}$.

**Lemma B.4.** *Sum of the weights in the groups, $\mathbf{u}_m \in \mathbb{Z}_+^{E_m}$, $\sum_{m=1}^{M} E_m = E$ has multinomial distribution:*

$$p(\{\mathbf{v}_1, ..., \mathbf{v}_M\}) = Mu(w, [\alpha_1, ..., \alpha_M]) $$
$$where: \alpha_m = \sum \boldsymbol{\theta}_m[i]. \tag{13}$$

*In the other words, the multinomial distribution is preserved when its counting variables are combined Siegrist (2017).*

**Theorem B.5.** *Given the aforementioned grouping of counts variables, the multinomial distribution can be modeled as a chain of binomials and multinomials:*

$$Mu(w, \boldsymbol{\theta} = [\boldsymbol{\theta}_1, ..., \boldsymbol{\theta}_M]) = \prod_{m=1}^{M} Bi(w - \sum_{i<m} \mathbf{v}_i, \eta_{\mathbf{v}_m}) \, Mu(\mathbf{v}_m, \boldsymbol{\lambda}_m), \tag{14}$$

$$where: \eta_{\mathbf{v}_m} = \frac{\mathbf{1}^T \boldsymbol{\theta}_m}{1 - \sum_{i<m} \mathbf{1}^T \boldsymbol{\theta}_i}, \ \boldsymbol{\lambda}_m = \frac{\boldsymbol{\theta}_m}{\mathbf{1}^T \boldsymbol{\theta}_m}$$

*Proof.* Since sum of the weights of the groups, $\mathbf{v}_m$, are functions of the weights in the group:

$$p(\mathbf{w}) = p(\mathbf{w}, \{\mathbf{v}_1, ..., \mathbf{v}_M\}) = p(\mathbf{w}|\{\mathbf{v}_1, ..., \mathbf{v}_M\})p(\{\mathbf{v}_1, ..., \mathbf{v}_M\})$$

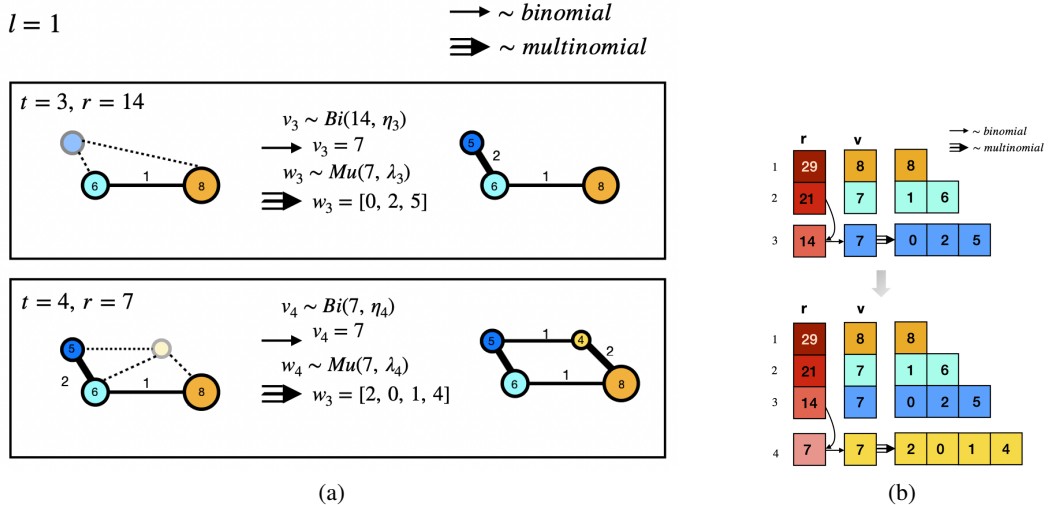

(a)          (b)

Figure 2: An illustration of the generation process of the single community in level $l = 1$ of $\mathcal{HG}$ in Figure 1a according to Theorem 3.2, and equation (5). The total weight of this community graph is 29, determined by the parent node of this community. Consequently, the edge probabilities of this community follow a Multinomial distribution. This Multinomial is formed as an autoregressive (AR) process and decomposed to a sequence of Binomials and Multinomials, as outlined in 3.2. At each iteration of this *stick-breaking* process, first a fraction of the remaining weights $r_t$ is allocated to the $t$-th row (corresponding to the $t$-th node in the sub-graph) and then this fraction, $v_t$, is distributed among that row of lower triangular adjacency matrix, $\hat{A}$.

According to lemma B.4, sum of the weights of the groups is a multinomial and by lemma B.3, it can be decomposed to a sequence of binomials:

$$p(\{v_1, ..., v_M\}) = \text{Mu}(w, \ [\alpha_1, ..., \alpha_M]) = \prod_{m=1}^{M} \text{Bi}(w - \sum\nolimits_{i<m} v_i, \hat{\eta}_m),$$

$$\text{where: } \alpha_m = \mathbf{1}^T \boldsymbol{\theta}_m, \ \hat{\eta}_e = \frac{\alpha_e}{1 - \sum_{i<e} \alpha_m}$$

Also based on lemma B.2, given the sum of the wights of all groups, the groups are independent and has multinomial distribution:

$$p(\mathbf{w}|\{v_1, ..., v_M\}) = \prod_{m=1}^{M} \text{Mu}(v_m, \ \boldsymbol{\lambda}_m)$$

$$\text{where: } \boldsymbol{\lambda}_m = \frac{\boldsymbol{\theta}_m}{\mathbf{1}^T \boldsymbol{\theta}_m}$$

$\square$

## B.4    TRAINING LOSS

According to equations (1) and (2), the log-likelihood for a graph sample can be written as

$$\log p(\mathcal{G}^L) = \sum_{l=1}^{L} \Big( \sum_{i \ \in \ \mathcal{V}(\mathcal{G}^{l-1})} \log p(\mathcal{C}_i^l \mid \mathcal{G}^{l-1}) + \sum_{(i,j) \in \ \mathcal{E}(\mathcal{G}^{l-1})} \log p(\mathcal{B}_{ij}^l \mid \mathcal{G}^{l-1}, \{\mathcal{C}_k^l\}_{\mathcal{C}_k^l \in \mathcal{G}^l}) \Big) \quad (15)$$

Therefore, one need to compute the log-likelihood of the communities and cross-community components. The log-likelihood of the cross-communities are straightforward using equation (7). For the

log-likelihood of the communities, as we break it into subsets of edges for each node in an autoregressive manner, with probability of each subset modeled in equation (5), therefore, the log-likelihood is reduced to

$$\log p(\mathcal{C}_i^l \mid \mathcal{G}^{l-1}) = \sum_{t=1}^{|\mathcal{C}_i^l|} \log p(\mathbf{u}_t(\hat{\mathcal{C}}_{i,t}^l))$$

where $p(\mathbf{u}_t(\hat{\mathcal{C}}_{i,t}^l))$ is defined in equation 5 as a mixture of product of binomial and multinomial. Since the binomial and multinomial are from the exponential family distribution, their log-likelihood reduces to Bregman divergence Banerjee et al. (2005). The binomial log likelihood is a general form of Bernoulli log likelihood, binary cross entropy, and multinomial log likelihood is a general form of Multinoulli (categorical) log likelihood, categorical cross entropy.

For training of generative model for the communities at level $l$ , we randomly sample total of $s$ augmented communities, so given $s_i$ of these sub-graphs are from community $\mathcal{C}_i^l$ then, we estimate the conditional generative probability for this community by averaging the loss function over all the subgraphs in that community multiplied by the size of the community:

$$\log p(\mathcal{C}_i^l \mid \mathcal{G}^{l-1}) = |\mathcal{C}_i^l| * mean([\log p(\mathbf{u}_t(\hat{\mathcal{C}}_{i,t}^l)), \ \forall \ t \in s_i]) \tag{16}$$

## C  MODEL ARCHITECTURE

### C.1  GRAPH NEURAL NETWORK (GNN) ARCHITECTURES

To overcome limitations in the sparse message passing mechanism, Graph Transformers (GTs) (Dwivedi & Bresson, 2020) have emerged as a recent solution. One key advantage of GTs is the ability for nodes to attend to all other nodes in a graph, known as global attention, which addresses issues such as over-smoothing, over-squashing, and expressiveness bounds Rampášek et al. (2022). GraphGPS provide a recipe for creating a more expressive and scalable graph transformer by making a hybrid message-passing graph neural networks (MPNN)+Transformer architecture. Additionally, recent GNN models propose to address the limitation of standard MPNNs in detecting simple substructures by adding features that they cannot capture on their own, such as the number of cycles. A framework for selecting and categorizing different types of positional and structural encodings, including *local, global, and relative* is provided in Rampášek et al. (2022). Positional encodings, such as eigenvectors of the adjacency or Laplacian matrices, aim to indicate the spatial position of a node within a graph, so nodes that are close to each other within a graph or subgraph should have similar positional encodings. On the other hand, structural encodings, such as degree of a node, number of k-cycles a node belong to or the diagonal of the $m$-steps random-walk matrix, aim to represent the structure of graphs or subgraphs, so nodes that share similar subgraphs or similar graphs should have similar structural encodings.

In order to encode the node features of the augmented graphs of bipartites, $\text{GNN}_{bp}^l(\hat{\mathcal{G}}^l)$, we customized GraphGPS in various ways. We incorporated distinct initial edge features to distinguish augmented (candidate) edges from real edges. Furthermore, for bipartite generation, we apply a mask on the attention scores of the transformers of the augmented graph $\hat{\mathcal{G}}^l$ to restrict attention only to connected communities. Specifically, the $i$-th row of the attention mask matrix is equal to 1 only for the index of the nodes that belong to the same community or the nodes of the neighboring communities that are linked by a bipartite, and 0 (i.e., no attention to those positions) otherwise.

The time and memory complexity of GraphGPS can be reduced to $\mathcal{O}(n + m)$ per layer by using *linear Transformers* such as Performer (Choromanski et al., 2020) or Exphormer, a sparse attention mechanism for graph, Shirzad et al. (2023) for global graph attention, while they can be as high quadratic in the number of nodes if the original Transformer architecture is employed. We leverage the original Transformer architecture for the experiments on all graphs datasets that are smaller than 1000 nodes except for point cloud dataset where Performer is used.

We also employed GraphGPS as the GNN of the parent graph, $\text{GNN}^{l-1}(\mathcal{G}^{l-1})$. On the other hand, for the community generation, we employed the GNN with attentive messages model, proposed in (Liao et al., 2019), as $\text{GNN}_{com}^l$. Additionally, we conducted experiments for the Enzyme dataset, using GraphGPS with the initial features as used in (Liao et al., 2019) for $\text{GNN}_{com}^l$ which resulted in comparable performance in the final graph generation.

## C.2 COMPLEXITY ANALYSIS

Given the linear complexity of $\mathcal{O}(n + m)$ for the Graph Neural Network (GNN) building blocks (GraphGPS and GAT), we can analyse the complexity of the proposed hierarchical graph generation. Let's denote the size of the largest community at level $l$ as $n_c^l := \max_i(|\mathcal{C}_i^l|)$ and $n_c := \max_{i,l}(|\mathcal{C}_i^L|)$. As explained in section 3, and illustrated in Figure 1 and algorithms (1, 2), each level $l$ of hierarchical generation is composed of:

0) Parent node embedding, $\text{GNN}^{l-1}(\mathcal{G}^{l-1})$, with $\mathcal{O}(n^{l-1} + m^{l-1})$.

1) Parallel community generation for $\{\mathcal{C}_i^l \; \forall i \in \mathcal{V}(\mathcal{G}^{l-1})\}$, which require $n_c^l$ generation steps. For each community $\hat{\mathcal{C}}_i^l$, node embedding computation, $\text{GNN}_{com}^l(\hat{\mathcal{C}}_i^l)$, requires $\mathcal{O}(n_i^l + m_i^l)$ operations, resulting in $n_c^l \sum_i \mathcal{O}(n_i^{l-1} + m_i^{l-1}) = n_c^l \mathcal{O}(n^l + m^l)$. In training all of these can be performed in parallel on the sampled batch.

2) Bipartite generation require $\mathcal{O}(n^l + \hat{m}^l)$ for node embedding computation, $\text{GNN}_{bp}^l(\hat{\mathcal{G}}^l)$, where $\hat{m}^l = |\mathcal{E}(\hat{\mathcal{G}}^l)| = \mathcal{O}(m^{l-1}(n_c^l)^2) \overset{(*)}{=} \mathcal{O}(n^l n_c^l)$. [5]

So, each level of graph generation requires $\mathcal{O}(n_c^l(n^l + m^l))$ computations, and consequently, the overall complexity of HiGen is $\sum_l^L \mathcal{O}(n_c^l(n^l + m^l)) = \mathcal{O}(n_c(n + m) \, L)$. Moreover, since most of the computations are parallelizable, graph sampling requires $\mathcal{O}(n_c \, L) = \mathcal{O}(n_c \log_{n_c} n)$ sequential steps to generate a graph of size $n$.

The complexity analysis for training follows a similar approach, but with the advantage that all steps can be executed in parallel. For batch training, we can adopt either subgraph-wise sampling or node-wise sampling, ensuring that each batch meets to GPU memory constraints (Duan et al., 2022). As detailed in Section B.4, the proposed model enables the random sampling of a total of $s$ augmented communities, eliminating the need to load the entire graph into memory, a distinction from diffusion models like DiGress.

**Complexity of partitioning algorithm:** Although optimizing modularity metric is an NP-hard problem in general, we used the Louvain algorithm, which is a greedy optimization method with $\mathcal{O}(n \log n)$ complexity. However, the graph partitioning is only applied once on the training data as a pre-processing step and its results are cached to be used throughout the entire training.

The pseudocodes for training and graph sampling using HiGen are presented in algorithms (1, 2).

Table 3: Complexity of different graph generative models. Here, $n$ is the number of nodes, $m$ is number of edges $n_c$ the size of the largest cluster graph and $L$ is the number of hierarchical levels. In the baseline models, $T$ is the number of diffusion steps, $K$ is the maximum number of active nodes during the diffusion process of EDGE (Chen et al., 2023).

| Model | Runtime | Sampling steps |
|---|---|---|
| **GraphRNN** | $\mathcal{O}(n^2)$ | $\mathcal{O}(n^2)$ |
| **GRAN** | $\mathcal{O}(n^2)$ | $\mathcal{O}(n)$ |
| **SPECTRE** | $\mathcal{O}(n^3)$ | $\mathcal{O}(1)$ |
| **GDSS** | $\mathcal{O}(T \, n^2)$ | $\mathcal{O}(T)$ |
| **DiGress** | $\mathcal{O}(T \, n^2)$ | $\mathcal{O}(T)$ |
| **EDGE** | $\mathcal{O}(T \, \max(m, K))$ | $\mathcal{O}(T)$ |
| **HiGen** | $\mathcal{O}(n_c(n + m) \, L)$ | $\mathcal{O}(n_c \, L)$ |

## C.3 CONNECTED GRAPH GENERATION

An advantage of the proposed model is its ability to enforce connected graph generation by constraining the total sum of the candidate edge weights to $\mathsf{v}_t > 0$ in the recursive *stick-breaking*

---

[5](*) We make an assumption that the graph is not very dense such that the number of edges is at the order of number of nodes, *i.e.* $m = \mathcal{O}(n)$.

---

**Algorithm 1** Training step of HiGen

---

1: **Input:** A hierarchical graph $\mathcal{HG} := \{\mathcal{G}^0, ...., \mathcal{G}^{L-1}, \mathcal{G}^L\}$
2: **for all** $l = 1$ to $L$ **do**                                                                     ▷ Can be done in parallel
3:     $\hat{\mathbb{C}}^l \leftarrow$ sample $s$ augmented communities $\hat{\mathcal{C}}^l_{i,t}$ from $\{\hat{\mathcal{C}}^l_{i,t} \mid i \le n^l_c,\ t \le |\mathcal{C}^l_i|\}$              ▷ $\mathcal{C}^l_i$ : $i$th community at level $l$
4:     $\hat{\mathcal{C}}^l \leftarrow \text{batch}(\hat{\mathbb{C}}^l)$
5:     $\hat{\mathcal{G}}^l \leftarrow \text{join}(\{\mathcal{C}^l_i\ \forall i \in \mathcal{V}(\mathcal{G}^{l-1})\}\ \cup\ \{\hat{\mathcal{B}}^l_{ij}\ \forall (i,j) \in \mathcal{E}(\mathcal{G}^{l-1})\})$ ▷ $\hat{\mathcal{B}}^l_{ij}$ : all the candidate edges of $\mathcal{B}^l_{ij}$ r.t. sec. 3.2.
6: **end for**
7: **for all** $l = 1$ to $L$ **do**                                                                     ▷ Can be done in parallel
8:     $h_{\mathcal{G}^{l-1}} \leftarrow \text{GNN}^{l-1}(\mathcal{G}^{l-1})$                             ▷ get parent node embeddings
9:     $h_{\hat{\mathcal{C}}^l} \leftarrow \text{GNN}^l_{com}(\hat{\mathcal{C}}^l)$
10:    $h_{\hat{\mathcal{G}}^l} \leftarrow \text{GNN}^l_{bp}(\hat{\mathcal{G}}^l)$                         ▷ skipped for $l = 1$ since it has no BP
11:    $loss^l \leftarrow \sum_{i\ \in\ \mathcal{V}(\mathcal{G}^{l-1})} \log p(\mathcal{C}^l_i \mid \mathcal{G}^{l-1}) + \sum_{(i,j)\in\ \mathcal{E}(\mathcal{G}^{l-1})} \log p(\mathcal{B}^l_{ij} \mid \mathcal{G}^{l-1})$     ▷ using eqn. (5), (16), (7)
12: **end for**
13: optimizer. $\text{step}(\sum^L_{l=1} loss^l)$

---

**Algorithm 2** Sampling from HiGen

---

1: $w_0 \sim p_{\mathbf{w}^0}(w_0)$                       ▷ $p_{\mathbf{w}^0}$ is the empirical distribution of the number of edge in training data
2:
3: **for** $l = 1$ to $L$ **do**
4:     $h_{\mathcal{G}^{l-1}} \leftarrow \text{GNN}^{l-1}(\mathcal{G}^{l-1})$                              ▷ 0) Get parent node embeddings
5:     $\hat{\mathbb{C}} \leftarrow \emptyset$                                                              ▽ 1) Generation of all communities
6:     **for all** $i = 1$ to $n^l_c = \mathcal{V}(\mathcal{G}^{l-1})$ **do**                               ▷ in parallel for all communities
7:         $\hat{\mathcal{C}}^l_i \leftarrow (\emptyset, \emptyset;\ r_i = w^{l-1}_{ii})$ ▷ Initialize with an empty graph and remaining edges' weight = the weight of the parent node
8:         $\hat{\mathbb{C}} \leftarrow \hat{\mathbb{C}} \cup \hat{\mathcal{C}}^l_i$
9:     **end for**
10:    **while** $\hat{\mathbb{C}} \neq \emptyset$ **do**                                                    ▷ grow all communities autoregressively
11:        $\hat{\mathcal{C}}^l \leftarrow \text{batch}(\hat{\mathbb{C}})$
12:        $h_{\hat{\mathcal{C}}^l} \leftarrow \text{GNN}^l_{com}(\hat{\mathcal{C}}^l)$
13:        $(\boldsymbol{\beta}^l, \eta^l_t, \boldsymbol{\lambda}^l_t) \leftarrow f(h_{\hat{\mathcal{C}}^l}, h_{\mathcal{G}^{l-1}})$                        ▷ using eqn. (6)
14:        **for all** $i = 1$ to $n^l_c = \mathcal{V}(\mathcal{G}^{l-1})$ **do**                            ▷ in parallel for all communities
15:            $k \sim Cat(\boldsymbol{\beta}^l_i)$                                                          ▷ sample mixture index from a categorical dist.
16:            $v \sim \text{Bi}(r_i,\ \eta^l_{t,k}[i])$                                                     ▷ sample $v$: sum of candidate edges for node (step) $t$
17:            $\boldsymbol{u} \sim \text{Mu}(v,\ \boldsymbol{\lambda}^l_{t,k}[i])$                          ▷ sample $\boldsymbol{u}$: weights of the candidate edges for node (step) $t$
18:            $r_i \leftarrow r_i - v$                                                                      ▷ update remaining edges' weight of $\hat{\mathcal{C}}^l_i$
19:            **if** $r_i = 0$ **then**                                                                     ▷ termination condition for generation of $\mathcal{C}^l_i$
20:                $\mathcal{C}^l_i \leftarrow \text{update}(\hat{\mathcal{C}}^l_i, \boldsymbol{u})$
21:                $\hat{\mathbb{C}} \leftarrow \hat{\mathbb{C}} \setminus \hat{\mathcal{C}}^l_i$
22:            **else**
23:                $\hat{\mathcal{C}}^l_i \leftarrow \text{update}(\hat{\mathcal{C}}^l_i, \boldsymbol{u}, r_i)$                      ▷ update $\hat{\mathcal{C}}^l_i$ with new sampled edges $\boldsymbol{u}$ for $t$th node
24:            **end if**
25:        **end for**
26:    **end while**
27:                                                                                                         ▽ 2) Bipartite generation (for $l \ge 2$)
28:    $\hat{\mathcal{G}}^l \leftarrow \text{join}(\{\mathcal{C}^l_i\ \forall i \in \mathcal{V}(\mathcal{G}^{l-1})\}\ \cup\ \{\hat{\mathcal{B}}^l_{ij}\ \forall (i,j) \in \mathcal{E}(\mathcal{G}^{l-1})\})$ ▷ $\hat{\mathcal{B}}^l_{ij}$ : all the candidate edges of $\mathcal{B}^l_{ij}$ r.t. sec. 3.2.
29:    $h_{\hat{\mathcal{G}}^l} \leftarrow \text{GNN}^l_{bp}(\hat{\mathcal{G}}^l)$
30:    $(\boldsymbol{\beta}^l, \boldsymbol{\theta}^l) \leftarrow f(h_{\hat{\mathcal{G}}^l}, h_{\mathcal{G}^{l-1}})$                             ▷ using eqn. (7)
31:    $k \sim Cat(\boldsymbol{\beta}^l_i)$                                                                  ▷ sample mixture index from a categorical dist.
32:    **for all** $(i,j) \in \mathcal{E}(\mathcal{G}^{l-1})$ **do**                                         ▷ in parallel for all BPs
33:        $\boldsymbol{w}^l_{ij} \sim \text{Mu}(w^{l-1}_{ij}, \boldsymbol{\theta}^l_{ij,k})$                 ▷ sample weights of $\mathcal{B}^l_{ij}$
34:    **end for**
35:    $\mathcal{G}^l \leftarrow \text{join}(\{\mathcal{C}^l_i\ \forall i \in \mathcal{V}(\mathcal{G}^{l-1})\}\ \cup\ \{\mathcal{B}^l_{ij}\ \forall(i,j) \in \mathcal{E}(\mathcal{G}^{l-1})\})$               ▷ Join all components
36: **end for**
37: **return** $\mathcal{G} = \mathcal{G}^L$

process. This is accomplished by modeling and sampling an auxiliary variable $\hat{v}_t \geq 0$ from the binomial distribution in theorem 3.2 as $Bi(\hat{v}_t \mid r_t - 1, \eta_{t,k}^l)$. Subsequently, its effective value is set as $v_t = \hat{v}_t + 1$, guaranteeing the existence of at least one edge among the candidate edges connecting the new node and the previously generated community in the autoregressive community generation process. This technique was particularly employed in our experiments with connected graphs.

It's noteworthy that the proposed method is not restricted to connected graphs and has the capability to model and generate graphs with disconnected components as well. HiGen, in essence, learns to reverse a graph coarsening algorithm, such as Louvain. In cases where a graph is disconnected, the coarsening algorithm produces an $\mathcal{HG}$ with a disconnected graph at the top level ($l = 1$ in Figure 1a). To illustrate, consider the example graph in Figure 1a is split into two left and right components (by removing the edge between the yellow and blue communities and the edge between the cyan and orange communities, and subsequently removing their corresponding edge at the top level ($l = 1$)), therefore the coarsened graph at the top level becomes a disconnected community composed of two components. Consequently, to potentially generate disconnected graphs, we would need to relax the community generation at level $l = 1$ to include disconnected communities. Meanwhile, we can maintain the constraint of connected community generation for the subsequent levels ($l > 1$).

### C.4 GENERATING GRAPH WITH NODE AND EDGE ATTRIBUTES

Adapting HiGen to handle attributed graphs involves reversing the partitioning (coarsening) algorithms tailored for clustering attributed graphs like molecular structures. For example a GNN-based clustering method inspired by the spectral relaxation of modularity metric for graphs with node attributes proposed by Tsitsulin et al. (2020). Toward that goal, we need to modify the proposed community generation and cross-community predictor to learn attributed edges or use the graph generation models with such capability.

Extending HiGen for graphs with edge types involves assigning a weight vector $\boldsymbol{w}_{i,j}^l \in \mathbb{Z}^d$ to each edge $e_{i,j}^l = (i, j)$, where $d$ is the number of edge types and each feature indicates the number of one specific edge type that the edge $e_{i,j}^l$ at a level $l$ is representing. The autoregressive multinomial and binomial generative models offered in this work can then be applied to each dimension so that the attributed graph at level $l + 1$ is generated based on its parent attributed graph $\mathcal{G}^l$. Note that, in this approach, the attribute of node $v_i$ can be added as a self loop edge $e_{i,i} = (i, i)$ with weight $\boldsymbol{w}_{i,i} = \boldsymbol{x}_i$. Consequently, the model only needs to predict the edge attributes (or edge types), aligning with HiGen's edge weight generation capabilities.

Another approach is training an additional predictor, such as a GNN model for edge/node classification, dedicated to predicting edge and node types based on the graph structure. This model does not have the difficulties associated with graph topology generation – such as graph isomorphism and edge independence – focusing solely on predicting edge/node attributes. This additional model can be tailored to specific application requirements and characteristics. Addressing attributed graphs is acknowledged as a potential future avenue for research.

## D EXPERIMENTAL DETAILS

**Datasets:** For the benchmark datasetst, graph sizes, denoted as $D_{dataset} = (|\mathcal{V}|_{max}, |\mathcal{V}|_{avg}, |\mathcal{E}|_{max}, |\mathcal{E}|_{avg})$, are: $D_{protein} = (500, 258, 1575, 646)$, $D_{Ego} = (399, 144, 1062, 332)$, $D_{Point-Cloud} = (5.03k, 1.4k, 10.9k, 3k)$,

Before training the models, we applied Louvain algorithm to obtain hierarchical graph structures for all of datasets and then trimmed out the intermediate levels to achieve uniform depth of $L = 2$. In case of $\mathcal{HG}$ s with varying heights, empty graphs can be added at the root levels of those $\mathcal{HG}$s with lower heights to avoid sampling them during training. Table 4 summarizes some statistics of the hierachical graph datasets. An 80%-20% split was randomly created for training and testing and 20% of the training data was used for validation purposes.

**Model Architecture:** In our experiments, the GraphGPS models consisted of 8 layers, while each level of hierarchical model has its own GNN parameters. The input node features were augmented with positional and structural encodings, which included the first 8 eigenvectors corresponding to

Table 4: Summary of some statistics of the benchmark graph datasets, Where $n_c = max(|\mathcal{C}|)$ denotes the size of largest cluster at the leaf level, $num_c$ is the number of clusters in each graph and $avg(mod_{test})$ and $avg(mod_{gen})$ are the modularity score of the test set and the generated samples by HiGen.

| dataset | $max(n)$ | $avg(n)$ | $avg(n_c)$ | $avg(num_c)$ | $avg(mod_{test})$ | $avg(mod_{gen})$ |
|---|---|---|---|---|---|---|
| **Enzyme** | 125 | 33 | 9.8 | 4.62 | 0.59 | 0.62 |
| **Ego** | 399 | 144 | 37.52 | 8.88 | 0.56 | 0.66 |
| **Protein** | 500 | 258 | 26.05 | 13.62 | 0.77 | 0.8 |
| **SBM** | 180 | 105 | 31.65 | 3.4 | 0.6 | 0.59 |
| **3D point Cloud** | 5K | 1.4K | 97.67 | 18.67 | 0.85 | 0.88 |

the smallest non-zero eigenvalues of the Laplacian matrices and the diagonal of the random-walk matrix up to 8 steps. We leverage the original Transformer architecture for all detests except Point Cloud dataset which use Performer. The hidden dimensions were set to 64 for the Protein, Ego, and Point Cloud datasets, and 128 for the Stochastic Block Model and Enzyme datasets. The number of mixtures was set to K=20.

In comparison, the GRAN models utilized 7 layers of GNNs with hidden dimensions of 128 for the Stochastic Block Model, Ego, and Enzyme datasets, 256 for the Point Cloud dataset, and 512 for the Protein dataset. Despite having smaller model sizes, HiGen achieved better performance than GRAN.

For training, the HiGen models used the Adam optimizer Kingma & Ba (2014) with a learning rate of 5e-4 and default settings for $\beta_1$ (0.9), $\beta_2$ (0.999), and $\epsilon$ (1e-8).

The experiments for the Enzyme and Stochastic Block Model datasets were conducted on a MacBook Air with an M2 processor and 16GB RAM, while the rest of the datasets were trained using an NVIDIA L4 Tensor Core GPU with 24GB RAM as an accelerator.

## D.1 MODEL SIZE COMPARISON

Here, we compare the model size, number of trainable parameters, against GRAN, the main baseline of our study. To ensure the HiGen model of the same size or smaller size than GRAN, we conducted the experiments for SBM and Enzyme datasets with reduced sizes. For the SBM dataset, we set the hidden dimension to 64, and for the Enzyme dataset, it was set to 32. The resulting model sizes and performance metrics are presented in Tables 5 and 6.

Table 5: Comparison of model sizes (number of trainable parameters) of HiGen vs GRAN.

| | **Protein** | **3D Point Cloud** | **Ego** | **Enzyme** | **SBM** |
|---|---|---|---|---|---|
| **GRAN** | 1.75e+7 | 5.7e+6 | 1.5e+7 | 1.54e+6 | 3.16e+6 |
| **HiGen** | 4.00e+6 | 6.26e+6 | 3.96e+6 | 1.48e+6 | 3.19e+6 |

Table 6: Comparison of generation metrics for Enzyme and Stochastic Block Model (SBM). Here, the hidden dimension of HiGen is set to 32 and 62 for Enzyme and SBM, respectively.

| Model | *Enzyme* Deg. ↓ | Clus. ↓ | Orbit ↓ |
|---|---|---|---|
| Training set | 0.0011 | 0.0025 | 3.7e-4 |
| GraphRNN | 0.017 | 0.062 | 0.046 |
| GRAN | 0.054 | 0.087 | 0.033 |
| GDSS | 0.026 | **0.061** | 0.009 |
| HiGen | **6.8**e-3 | 0.067 | **1.3**e-3 |

| Model | *Stochastic block model* Deg. ↓ | Clus. ↓ | Orbit↓ | Spec. ↓ |
|---|---|---|---|---|
| Training set | 0.0008 | 0.0332 | 0.0255 | 0.0063 |
| GraphRNN | 0.0055 | 0.0584 | 0.0785 | 0.0065 |
| GRAN | 0.0113 | 0.0553 | 0.0540 | 0.0054 |
| SPECTRE | 0.0015 | 0.0521 | 0.0412 | 0.0056 |
| DiGress | **0.0013** | **0.0498** | 0.0433 | - |
| HiGen | 0.0015 | 0.0520 | **0.0370** | **0.0049** |

These results highlight that despite smaller or equal model sizes, HiGen outperforms GRAN's performance. This emphasizes the efficacy of hierarchically modeling communities and cross-community interactions as distinct entities. This results can be explained by the fact that the proposed

model needs to learn smaller community sizes compared to GRAN, allowing for the utilization of more compact models.

## D.2 Sampling Speed Comparison

In table 8, we present a comparison of the sampling times between the proposed method and its primary counterpart, GRAN, measured in seconds. The sampling processes were carried out on a server machine equipped with a 32-core AMD Rome 7532 CPU and 128 GB of RAM.

Table 7: Sampling times in seconds.

|        | Protein | Ego   | SBM    | Enzyme |
|--------|---------|-------|--------|--------|
| **GRAN**  | 46.04   | 2.145 | 1.5873 | 0.2475 |
| **HiGen** | 1.33    | 0.528 | 0.4653 | 0.1452 |

Table 8: Sampling speedup factor of generative model vs GRAN: $t_{GRAN}(s)/t_{model}(s)$. The speedup factor of baseline models are obtained from Martinkus et al. (2022).

|              | Protein | SBM   |
|--------------|---------|-------|
| **GRAN**     | 1       | 1     |
| **GraphRNN** | 0.32    | 0.37  |
| **SPECTRE**  | 23.04   | 25.54 |
| **HiGen**    | 34.62   | 3.41  |

As expected by the model architecture and is evident from the table 8, HiGen demonstrates a significantly faster sampling, particularly for larger graph samples.

## E Additional Results

Table 9 presents the results of various metrics for HiGen models on all benchmark datasets. The structural statistics are evaluated using the Total Variation kernel as the Maximum Mean Discrepancy (MMD) metric.

In addition, the table includes the average of random-GNN-based metrics (Thompson et al., 2022) over 10 random Graph Isomorphism Network (GIN) initializations. The reported metrics are MMD with RBF kernel (GNN RBF), the harmonic mean of improved precision+recall (GNN F1 PR) and harmonic mean of density+coverage (GNN F1 PR).

Table 9: Various graph generative performance metrics for HiGen models on all benchmark datasets.

| Model | Deg. ↓ | Clus. ↓ | Orbit↓ | Spec. ↓ | GNN RBF ↓ | GNN F1 PR ↑ | GNN F1 DC ↑ |
|---|---|---|---|---|---|---|---|
| *Enzyme* | | | | | | | |
| **GRAN** | 8.45e-03 | 2.62e-02 | 2.11e-02 | 3.46e-02 | 0.0663 | 0.950 | 0.832 |
| **HiGen-m** | 6.61e-03 | 2.65e-02 | 2.15e-03 | 8.75e-03 | 0.0215 | 0.970 | 0.897 |
| **HiGen** | 2.31e-03 | 2.08e-02 | 1.51e-03 | 9.56e-03 | 0.0180 | 0.978 | 0.983 |
| *Protein* | | | | | | | |
| **HiGen-m** | 0.0041 | 0.109 | 0.0472 | 0.0061 | 0.167 | 0.912 | 0.826 |
| **HiGen** | 0.0012 | 0.0435 | 0.0234 | 0.0025 | 0.0671 | 0.979 | 0.985 |
| *Stochastic block model* | | | | | | | |
| **GRAN** | 0.0159 | 0.0518 | 0.0462 | 0.0104 | 0.0653 | 0.977 | 0.86 |
| **HiGen-m** | 0.0017 | 0.0503 | 0.0604 | 0.0068 | 0.154 | 0.912 | 0.83 |
| **HiGen** | 0.0019 | 0.0498 | 0.0352 | 0.0046 | 0.0432 | 0.986 | 1.07 |
| *Ego* | | | | | | | |
| **GraphRNN** | 9.55e-3 | 0.094 | 0.048 | 0.025 | 0.0972 | 0.86 | 0.45 |
| **GRAN** | 7.65e-3 | 0.066 | 0.043 | 0.026 | 0.0700 | 0.76 | 0.50 |
| **HiGen-m** | 0.011 | 0.063 | 0.021 | 0.013 | 0.0420 | 0.87 | 0.68 |
| **HiGen** | 1.9e-3 | 0.049 | 0.029 | 0.004 | 0.0520 | 0.88 | 0.69 |

Table 10: Performance comparison of HiGen model on Ego datasets against the baselines reported in (Chen et al., 2023). Gaussian EMD kernel was used for structure-based statistics together with GNN-based performance metrics: GNN RBF and Frechet Distance (FD)

| | *Ego* | | | | |
|---|---|---|---|---|---|
| Model | Deg. | Clus. | Orbit | GNN RBF | FD |
| **GraphRNN** | 0.0768 | 1.1456 | 0.1087 | 0.6827 | 90.57 |
| **GRAN** | 0.5778 | 0.3360 | 0.0406 | 0.2633 | 489.96 |
| **GDSS** | 0.8189 | 0.6032 | 0.3315 | 0.4331 | 60.61 |
| **DiscDDPM** | 0.4613 | 0.1681 | 0.0633 | 0.1561 | 42.80 |
| **DiGress** | 0.0708 | 0.0092 | 0.1205 | 0.0489 | 18.68 |
| **EDGE** | 0.0579 | 0.1773 | 0.0519 | 0.0658 | 15.76 |
| **HiGen-m** | 0.114 | 0.0378 | 0.0535 | **0.0420** | 12.2 |
| **HiGen** | **0.0472** | **0.0031** | **0.0387** | 0.0454 | **5.24** |

As the results show, HiGen outperforms other baseline, particularly improving in terms of degree statistic compared to EDGE which is explicitly a degree-guided graph generative model by design.

## E.1 POINT CLOUD

We also evaluated HiGen on the *Point Cloud* dataset, which consists of 41 simulated 3D point clouds of household objects. This dataset consists of large graphs of approximately 1.4k nodes on average with maximum of over 5k nodes. In this dataset, each point is mapped to a node in a graph, and edges are connecting the k-nearest neighbors based on Euclidean distance in 3D space (Neumann et al., 2013). We conducted the experiments for hierarchical depth of $L = 2$ and $L = 3$. In the case of $L = 2$, we spliced out the intermediate levels of Louvain algorithm's output. For the $L = 3$ experiments, we splice out all the intermediate levels except for the one that is two level above the leaf level when partitioning function's output had more than 3 levels to achieve final $\mathcal{HG}$s with uniform depth of $L = 3$. For all graphs, Louvain's output had at least 3 levels.

However, the quadratic growth of the number of candidate edges in the augmented graph of bipartites $\hat{\mathcal{G}}^l$ – the graph composed of all the communities and the candidate edges of all bipartites used in section 3.2 for bipartite generation – can cause out of memory issue when training large graphs in the point cloud dataset. To address this issue, we can sample sub-graphs and generate one (or a subset of) bipartites at a time to fit the available memory. In our experimental study, we generated bipartites sequentially, sorting them based on the index of their parent edges in the parent level. In this case, the augmented graph $\hat{\mathcal{G}}^l$ used for generating the edges of $\mathcal{B}_{ij}^l$ consists of all the communities

$\{\mathcal{C}_k^l \;\; \forall k \le \max(i,j)\}$ and all the bipartites $\{\mathcal{B}_{mn}^l \;\; \forall(m,n) \le (i,j)\}$, augmented with the candidate edges of $\mathcal{B}_{ij}^l$. This model is denoted by HiGen-s in table 11.

This modification can also address a potential limitation related to edge independence when generating all the inter-communities simultaneously. However, it's important to note that the significance of edge independence is more prominent in high-density graphs like community generations, (Chanpuriya et al., 2021), whereas its impact is less significant in sparser inter-communities of hierarchical approach. This is evident by the performance improvement observed in our experiments.

The GraphGPS models that was used for this experiment have employed Performer (Choromanski et al., 2020) which offers linear time and memory complexity. The results in Table 11 highlights the performance improvement of HiGen-s in both local and global properties of the generated graphs.

Table 11: Comparison of generation metrics on benchmark 3D point cloud for $L = 2$ and deeper hierarchical model of $L = 3$. The baseline results are obtained from (Liao et al., 2019).

|  | 3D Point Cloud | | | |
| Model | Deg. ↓ | Clus. ↓ | Orbit↓ | Spec.↓ |
|---|---|---|---|---|
| **Erdos-Renyi** | 3.1e-01 | 1.22 | 1.27 | 4.26e-02 |
| **GRAN** | **1.75e-02** | 5.1e-01 | 2.1e-01 | 7.45e-03 |
| **HiGen-s (L=2)** | 3.48e-02 | **2.82e-01** | 3.45e-02 | 5.46e-03 |
| **HiGen-s (L=3)** | 4.97e-02 | 3.19e-01 | **1.97e-02** | **5.2e-03** |

An alternative approach is to sub-sample a large graph such that each augmented sub-graph consists of a bipartite $\mathcal{B}_{ij}^l$ and its corresponding pair of communities $\mathcal{C}_i^l$, $\mathcal{C}_j^l$. This approach allows for parallel generation of bipartite sub-graphs on multiple workers but does not consider the edge dependece between neighboring bipartites.

## E.2 ABLATION STUDIES

In this section, two ablation studies were conducted to evaluate the sensitivity of HiGen with different node orderings and graph partitioning functions.

**Node Ordering** In our experimental study, the nodes in the communities of all levels are ordered using breadth first search (BFS) node ordering while the BFS queue are sorted by the total weight of edges between a node in the queue and predecessor nodes plus its self-edge. To compare the sensitivity of the HiGen model against GRAN versus different node orderings, we trained the models with default node ordering and random node ordering. The performance results, presented in Table 12, confirm that the proposed model is significantly less sensitive to the node ordering whereas the performance of GRAN drops considerably with non-optimal orderings.

Table 12: Ablation study on node ordering. Baseline HiGen used the BFS ordering and baseline GRAN used DFS ordering. $\pi_1$, $\pi_2$ and $\pi_3$ are default, random and $\pi_3$ node ordering, respectively. Total variation kernel is used as MMD metrics of structural statistics. Also, the average of random-GNN-based metrics aver 10 random GIN initialization are reported for MMD with RBF kernel (GNN RBF), the harmonic mean of improved precision+recall (GNN F1 PR) and harmonic mean of density+coverage (GNN F1 PR).

|  | Enzyme | | | | | | |
| Model | Deg. ↓ | Clus. ↓ | Orbit↓ | Spec.↓ | GNN RBF ↓ | GNN F1 PR ↑ | GNN F1 DC ↑ |
|---|---|---|---|---|---|---|---|
| **GRAN** | 8.45e-03 | 2.62e-02 | 3.46e-02 | 2.11e-02 | 6.63e-02 | 9.50e-01 | 8.32e-01 |
| **GRAN ($\pi_1$)** | 1.75e-02 | 2.89e-02 | 3.78e-02 | 2.03e-02 | 6.51e-02 | 8.24e-01 | 6.69e-01 |
| **GRAN ($\pi_2$)** | 3.90e-02 | 3.24e-02 | 3.81e-02 | 2.38e-02 | 1.26e-01 | 8.31e-01 | 6.72e-01 |
| **HiGen** | 2.31e-03 | 2.08e-02 | 1.51e-03 | 9.56e-03 | 1.80e-02 | 9.78e-01 | 9.83e-01 |
| **HiGen ($\pi_1$)** | 1.83e-03 | 2.21e-02 | 6.75e-04 | 7.08e-03 | 1.78e-02 | 9.84e-01 | 9.77e-01 |
| **HiGen ($\pi_2$)** | 3.31e-03 | 2.34e-02 | 2.06e-03 | 9.10e-03 | 2.04e-02 | 9.47e-01 | 8.81e-01 |
| **HiGen ($\pi_3$)** | 1.34e-03 | 2.13e-02 | 6.94e-04 | 6.56e-03 | 1.90e-02 | 9.61e-01 | 9.74e-01 |

**Different Graph Partitioning**    In this experimental study, we evaluated the performance of HiGen using different graph partitioning functions. Firstly, to assess the sensitivity of the hierarchical generative model to random initialization in the Louvain algorithm, we conducted the HiGen experiment three times with different random seeds on the Enzyme dataset. The average and standard deviation of performance metrics are reported in Table 13 which demonstrate that HiGen consistently achieves almost similar performance across different random initializations.

Additionally, we explored spectral clustering (SC), which is a relaxed formulation of $k$-min-cut partitioning (Shi & Malik, 2000), as an alternative partitioning method. To determine the number of clusters, we applied SC to partition the graphs over a range of $0.7\sqrt{n} \leq k \leq 1.3\sqrt{n}$, where $n$ represents the number of nodes in the graph. We computed the modularity score of each partition and selected the value of $k$ that yielded the maximum score.

The results presented in Table 13 demonstrate the robustness of HiGen against different graph partitioning functions.

Table 13: Multiple initialization of Louvain partitioning algorithm and also min-cut partitioning

| Model | Deg. ↓ | Clus. ↓ | Orbit↓ | *Enzyme*
Spec.↓ | GNN RBF ↓ | GNN F1 PR ↑ | GNN F1 DC ↑ |
|---|---|---|---|---|---|---|---|
| **HiGen** | 2.64e-03±4.7e-4 | 2.09e-02±4.0e-4 | 7.46e-04±4.4e-4 | 1.74e-02±1.5e-3 | 2.00e-02±3.1e-3 | .98±4.6e-3 | .96±1.0e-2 |
| **HiGen (SC)** | 2.24e-03 | 2.10e-02 | 5.59e-04 | 8.30e-03 | 2.00e-02 | .98 | .94 , |

### E.3    GRAPH SAMPLES

Generated hierarchical graphs sampled from HiGen models are presented in this section.

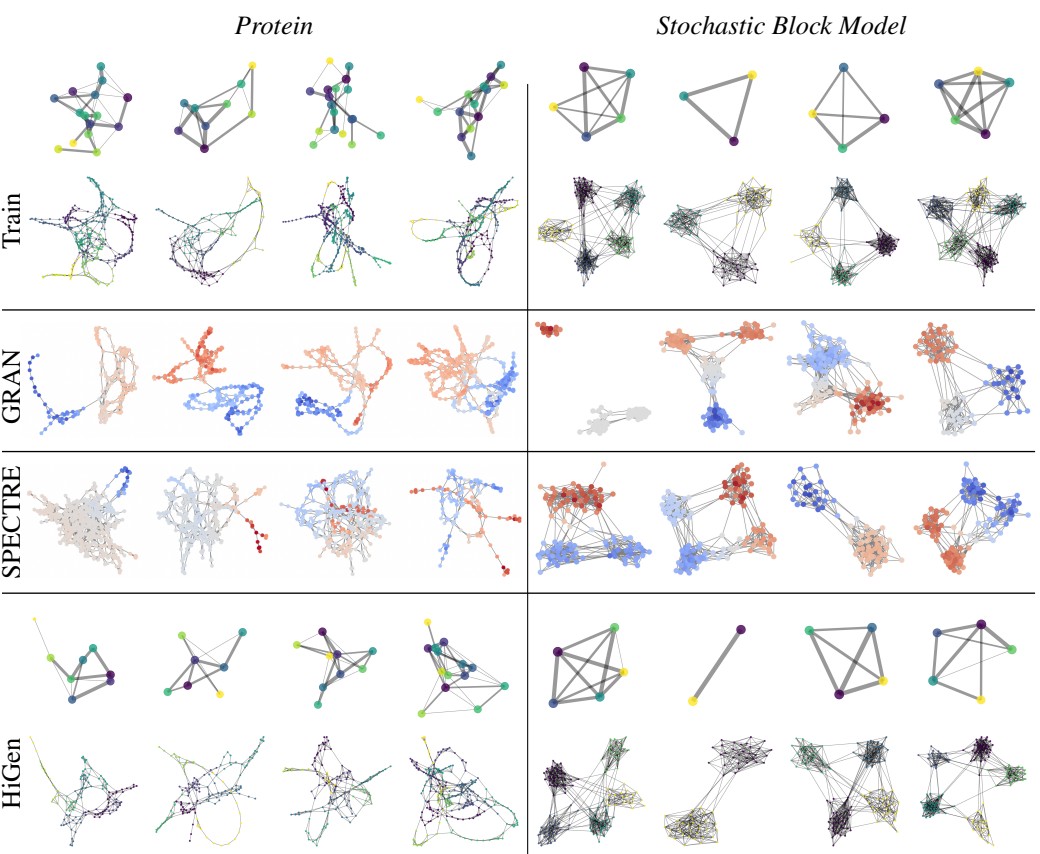

Figure 3: Samples from HiGen trained on *Protein* and *SBM*. Communities are distinguished with different colors and both levels are depicted. The samples for GRAN and SPECRE are obtained from (Martinkus et al., 2022).

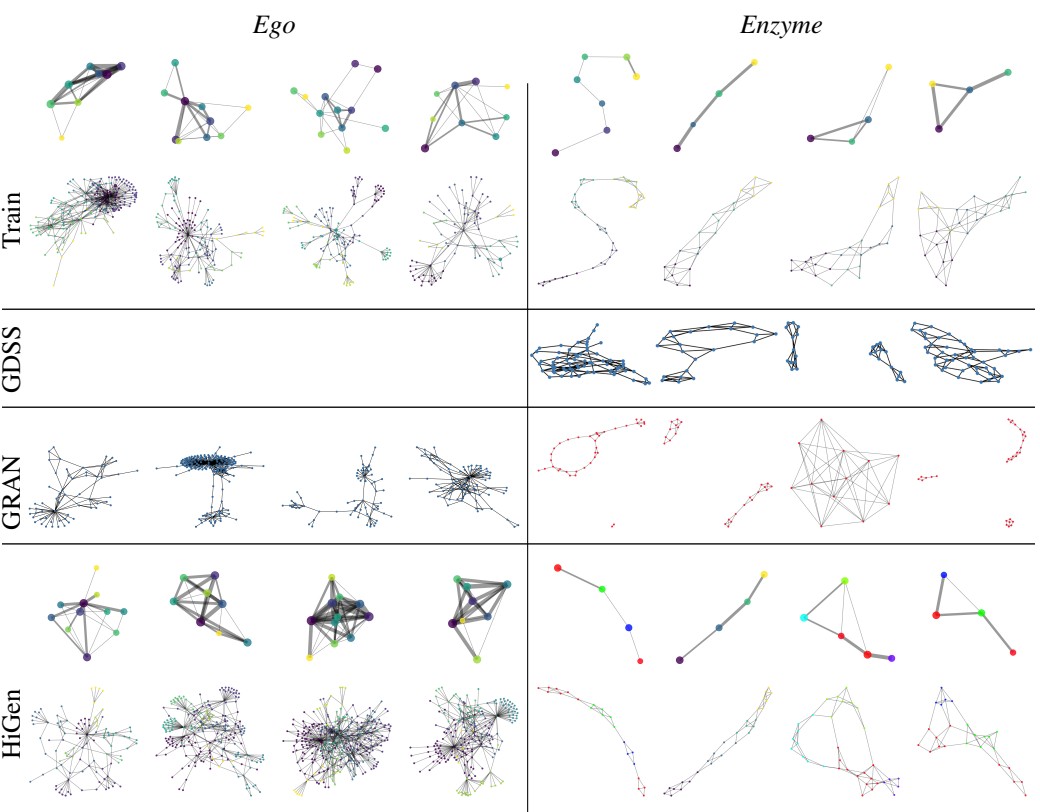

Figure 4: Samples from HiGen trained on *Protein* and *SBM*. Communities are distinguished with different colors and both levels are depicted. The samples for GDSS are obtained from (Jo et al., 2022).

*3D Point Cloud*

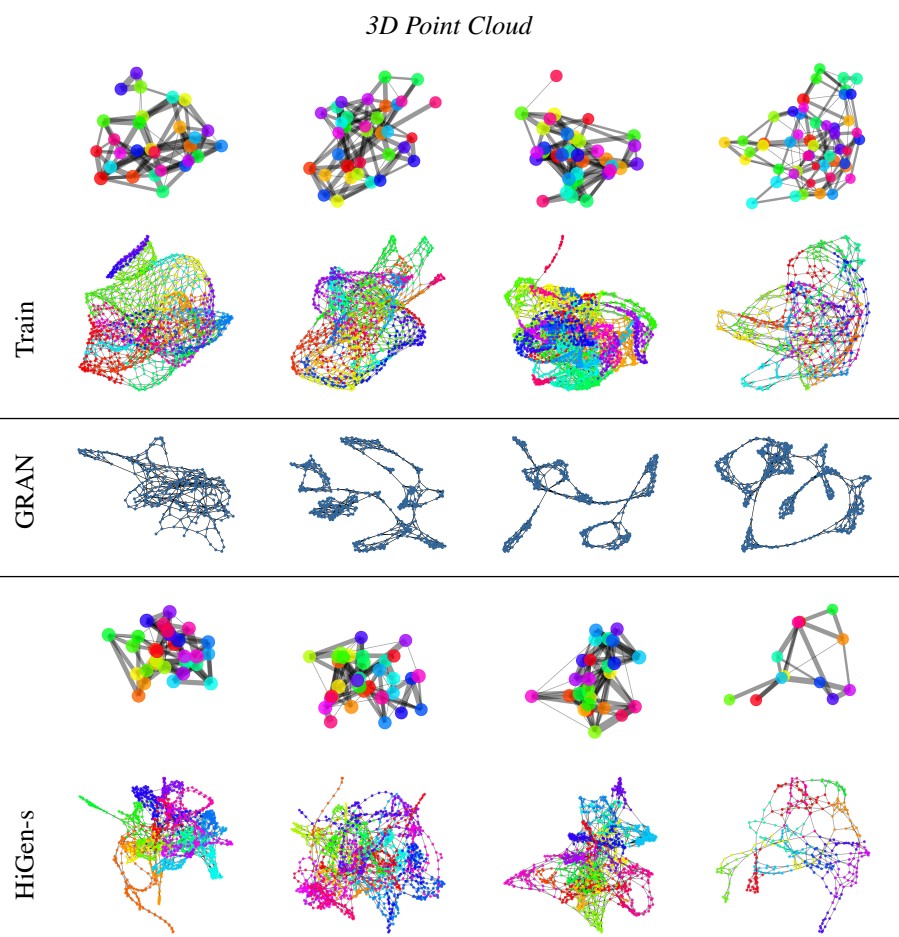

Figure 5: Samples from HiGen trained on *3D Point Cloud*. Communities are distinguished with different colors and both levels are depicted. The samples for GRAN are obtained from (Liao et al., 2019).

