# OpenReview forum: "HiGen: Hierarchical Graph Generative Networks"
_ICLR.cc/2024/Conference — ICLR 2024 poster_

### Official Review · Reviewer_WR6B · 2023-10-26

**Soundness:** 3 good
**Presentation:** 3 good
**Contribution:** 2 fair
**Rating:** 6
**Confidence:** 3

**Summary:**

The authors propose a new deep graph generative model, HiGen. HiGen generates a graph in a coarse-to-fine manner, wherein a small, "low-resolution" graph is generated, and at the next, higher-resolution level, each node at the prior level corresponds to the graph of a community of nodes, whereas each edge corresponds to a bipartite graph between two communities. The concept of this method promises greater scalability, parallelization, and graph quality than prior methods. Experiments on generating several classes of graphs, such as SBMs, proteins, enzymes, ego networks, and point clouds, indicate overall superiority of the quality of graphs produced by HiGen relative to some prior methods.

**Strengths:**

1) The introduction features an informative summary of recent work in graph generative deep networks. It positions the author's method well in the context of this prior work.
2) The topic of the paper, deep graph generation, is very popular in recent years.
3) The concept of the method is logical and promises better runtime than some prior methods. Experiments indicate better graph quality than prior methods as well.

**Weaknesses:**

1) There is limited theoretical advancement. The theorems in this work regard the correctness of an aspect of the graph generation (specifically, the correctness of a certain factorization of the multinomial distribution). The introduction alludes to challenges in graph generation like "difficulty capturing complex dependencies in graph structures," learning multi-scale structure, etc., but there is no theory addressing how well the proposed algorithm performs at this task relative to others.
2) Training / sampling from HiGen could be broken out into algorithms in the text for clearer presentation. At present, reviewing what the algorithms are requires going through several pages of text.

Typos:
- page 1: Jin et al. unparenthesized
- page 2: proposed *a* generative model
- page 7: Kong et al. unparenthesized
- page 7: "an analytically solution"
- page 7: "n our experiments,"
- page 8: "However, It’s important"

**Questions:**

1) As stated above, I suggest breaking out the training/sampling into algorithms to improve readability.

2) There is an abundance of papers proposing new deep graph generative models in recent years, as outlined in this paper's introduction, and these papers generally claim superior graph quality to prior methods. However, there are many degrees of freedom in measuring the quality of graph generation, so it is hard to tell whether there is real progress. With this in mind, how would the authors argue that there is a real advancement in the graph quality of HiGen? As I mentioned above, a theoretical framework is one possibility, but this paper goes in a more empirical direction.

---

> ### Author Response · Authors · 2023-11-17
>
> Dear reviewer,
> We appreciate your valuable feedback and review. Here are our responses to your concerns
> 1. We have incorporated training and sampling pseudocode into the appendix, complementing the visual representations in Figures 1 and 2 to illustrate and improve the readability of the proposed method.
>
> 2.a Our evaluation methodology aligns with established practices in the literature for comparing generated graph quality, comparing not only widely accepted metrics such as MMD for structure-based statistics against the best results reported in the baseline papers but also evaluating and reporting the recent GNN-based performance metrics. The reported improvements in graph generation quality, together with enhanced sampling time and scalability to large point cloud graphs, where many existing methods face challenges, signify a **substantial advancement**. Complexity and sampling time analysis also supports the scalability of the proposed method.
>
> 2.b **Theoretical Framework:** The proposed method is supported by theoretical results that contribute to its effectiveness. The hierarchical chain structure for graph generation (eq. (1)), the use of separate networks for modeling communities and bipartites, the application of multinomial models (as demonstrated in Theorem 3.1), and decomposition of the multinomial distributions as a recursive model (Theorem 3.2 and Lemma B.3), all form part of a robust theoretical foundation.  The focus on multinomial distribution for edge weight is justified by the need to reverse a graph coarsening algorithm that preserves the number of edges in the community and cross-community in the coarsened hierarchical graph. The weights in such a hierarchical graph can best be described by a multinomial distribution (more details are presented in section 3 and lemma B.2). Since the multinomial is a joint distribution over all the edges in the graph, in order to get a scalable and efficient generative model it is crucial to break it into distribution of the sub-graphs (as presented by factorization results in Theorem 3.1) and also to describe this joint distribution as an autoregressive generative model (Theorem 3.2 for group-wise autoregressive generation or Lemma B.3 for edgewise autoregressive generation). Moreover, the term "learning multi-scale" in the introduction and the conclusion refers to the hierarchical and cluster-based generative model inherent in our approach.
>
> **Typos:** are fixed in the new version.
>
> We hope these have addressed your questions. Your insights and suggestions are highly valued and we remain committed to addressing any outstanding concerns you may have.

---

> > ### Comment · Reviewer_WR6B · 2023-11-21
> >
> > Thank you for your response and revisions. The addition of the pseudocode has mostly alleviated my concerns regarding the presentation. Regarding the evaluation, I was not questioning whether the paper follows these accepted metrics, but rather whether only improving on these metrics is necessarily productive, as opposed to also providing theoretical or applied contributions. This is not a strong demerit against the paper, since it applies to many recent papers in this area. Overall, I think the revisions have enhanced the paper, and I have raised my score.

---

### Official Review · Reviewer_oTgM · 2023-10-29

**Soundness:** 3 good
**Presentation:** 3 good
**Contribution:** 3 good
**Rating:** 6
**Confidence:** 3

**Summary:**

The paper proposes a general hierarchical graph generation method, aiming to generate graph in a coarser-to-fine way. The proposed idea is quite reasonable for graph data. The proposed method learns the probability of connectivity of communities and the edges in each community conditioned on the graph of privious layer. Extensive results on several kinds of datasets well demonstrate the effectiveness of the proposed method that could generate graphs with desired properties.

**Strengths:**

1. The proposed method is novel, where the idea of generating graphs without prior knowledge has not been studied yet. And the idea is also reasonable for generating various kinds of real-world graphs.
2. The proposed method is sound.
3. The experiments validate the proposed method could generate a hierarchical graph structure.

**Weaknesses:**

1. The computation complexity of the proposed model is not clear.
2. Could the proposed method be applied to molecule generation and compared with HierVAE? (Jin et al. 2020)
3. How to identify the number of layers and the number of communities in each layer?
4. The metric of novelty is also important for graph generation method. How about the novelty of the generated graphs by the proposed method?

**Questions:**

Please refer to the weaknesses.

---

> ### Author Response · Authors · 2023-11-17
>
> Dear reviewer,
>
> Thank you for your thorough review and the insightful feedback. We are pleased to see that you found the proposed hierarchical graph generative model compelling and recognized its novelty. Here are our responses to your questions:
> 1. We have included a **complete complexity analysis** together with pseudocodes for training and graph sampling in the revised version of the manuscript.
> 2. **Molecule graph generation**: HiGen offers a versatile framework for efficient and scalable graph generation,  emphasizing on capturing the hierarchical and community structures in graphs. In order to generate attributed graphs, such as molecules,  we need to modify the proposed community generation and cross-community predictor to learn attributed edges or use the graph generation models with such capability.
>
> One strategy for extending HiGen for graphs with edge types involves assigning a weight vector $\mathbf{w}\_{i, j}^l \in \mathbb{Z}^d$ to each edge $e\_{i, j}^l = (i, j)$, where $d$ is the number of edge types and each feature indicates the number of one specific edge type that the edge $e\_{i,j}^l$ at a level $l$ is representing. The autoregressive multinomial and binomial generative models offered in this work can then be applied to each dimension so that the attributed graph at level $l+1$ is generated based on its parent attributed graph $\mathcal{G}^l$. Note that, in this approach, the attribute of node $v_i$ can be added as a self loop edge $e\_{i,i} = (i, i)$ with weight $\mathbf{w}\_{i,i}=\mathbf{x}\_i$.
>
> Another approach is training an additional predictor, such as a GNN model for edge/node classification, dedicated to predicting edge and node types based on the graph structure. We incorporated a detailed discussion for attributed graph generation  in Appendix C.
>
> 3. In our experiments, we predefined the **number of levels**. However, the flexibility of this generative model extends to datasets with a varying number of levels, for which we can augment $\mathcal{HG}$s with fewer levels by appending empty graphs at the top levels during training, ensuring these levels are not sampled. During generation, we can sample the number of levels from the empirical distribution (histogram) observed in the training graphs. Currently, we utilize the empirical distribution of the training dataset solely to estimate the total number of edges $w_0$, implicitly indicating the final size of the graph.
>
> Regarding the **number of communities** at level $l$, it corresponds to the number of nodes at the parent level. Each node at level $l-1$ becomes a community in the subsequent level.
>
> **Community sizes**: It is worth mentioning that a notable feature of the proposed multinomial autoregressive (AR) model is that it does not require an extra model to estimate the community sizes before generation. At the initiation of the process for generating community $\mathcal{C}\_i$, the total and remaining weight of the community are set based on its parent node weight, ($r_i = w\_{ii}^{l-1}$). progressively adds new nodes and samples their edges using the recursive multinomial/binomial process outlined in Theorem 3.2 until the remaining weight reaches $0$.  Further details are provided in Algorithm 2 in Appendix C.
>
> 4.  Although **Novelty metric** was used in (Martinkus et al., 2022) to ensure diverse graph generation, it doesn’t seem to serve as a distinctive metric as it was reported as 100% across all the generative algorithms for SBM and Protein datasets in that study.
>
> In contrast, random-GNN-based metrics such as GNN RBF (using MMD with RBF kernel) and GNN F1 PR (harmonic mean of improved precision+recall), that we reported in table 1 for Ego dataset and appendix E, exhibit strong correlation with the graph diversity (Thompson et al., 2022).
>
> We hope our response has addressed all your questions. We highly value your insights and suggestions and are committed to addressing any outstanding concern you may have.

---

### Official Review · Reviewer_tztH · 2023-10-30

**Soundness:** 3 good
**Presentation:** 3 good
**Contribution:** 3 good
**Rating:** 6
**Confidence:** 3

**Summary:**

This paper proposes HiGen, a novel graph generative network that captures the hierarchical nature of graphs and successively generates the graph sub-structures in a coarse-to-fine fashion. This method enables scalable graph generation for large and complex graphs, while generating community graphs with integer-valued edge weights in an autoregressive manner. Empirical studies demonstrate the effectiveness and scalability of the proposed method, achieving state-of- the-art performance in terms of graph quality across various benchmark datasets.

**Strengths:**

- The idea of generating real-world graphs hierarchically is novel and interesting.

- The resulting HiGen model can generate high-quality real-world graphs, with theoretical support on community generation.

- The experiments are convincing, showing that HiGen can outperform many graph generation models on a wide range of datasets.

**Weaknesses:**

- The authors should include a complete analysis of complexity against previous methods, including the complexity of graph partitioning.

**Questions:**

- See weakness above.
- Typo: Section 5, paragraph “Model Architecture”

---

> ### Author Response · Authors · 2023-11-17
>
> Dear reviewer,
>
> Thank you for your thorough review and the insightful feedback. We are pleased to see that you found the proposed method compelling and recognized its strengths. In response to your suggestion, we have included a complete complexity analysis together with pseudocodes for training and graph sampling in the revised version of the manuscript. We hope this addition addresses your concern. Your feedback is highly valuable and we are committed to addressing any outstanding concern you may have.

---

### Official Review · Reviewer_D2iQ · 2023-11-01

**Soundness:** 3 good
**Presentation:** 3 good
**Contribution:** 3 good
**Rating:** 6
**Confidence:** 3

**Summary:**

In this paper, the authors introduce Hierarchical Graph Generative Networks (HIGEN), a model designed to encapsulate the hierarchical characteristics of graphs through a progressive generation of graph sub-structures, transitioning from broader to more detailed aspects. At every hierarchical level, the model concurrently produces communities, subsequently generating bipartite graphs to represent cross-edges between these communities, utilizing distinct neural networks for each task. This compartmentalized strategy ensures that the graph generation process is both scalable and efficient, even when applied to large and intricate graphs. The method presented surpasses the performance of current leading techniques across a range of benchmark datasets.

**Strengths:**

1. HIGEN adeptly grasps the hierarchical nature of real-world graphs, facilitating the generation of sub-structures in a manner that is both scalable and efficient.

2. The authors conduct a comprehensive evaluation of the proposed method, utilizing a variety of benchmark datasets to showcase the method's capability in accurately generating graphs that reflect the statistical characteristics inherent to real-world graphs.

3. The manuscript offers an in-depth examination of the graphs produced by the HIGEN models, including a visual comparison of these generated graphs and a rigorous experimental assessment of diverse node ordering and partitioning functions.

4. The authors also present an analysis of computational complexity, alongside a comparison of sampling speeds, providing a holistic understanding of the method's performance and efficiency.

**Weaknesses:**

1. The proposed method assumes that the input graph is connected, which may not be the case for some real-world graphs.
2. Due to its hierarchical generation approach, particularly during the community generation phase, the proposed method might face challenges in maintaining control over the global distribution of the graph.

Minor Problem:
Typo: "Model Architecture n our experiments"

**Questions:**

Please refer to weaknesses.

---

> ### Author Response · Authors · 2023-11-16
>
> Dear reviewer,
>
> Thank you for your thorough review and the insightful and constructive feedback. We are pleased to see that you found the proposed method compelling and recognized its strengths. In the following, we answer your questions:
> 1. **Connected Graph Assumption:** Indeed, the proposed method is not restricted to connected graphs and has the capability to model and generate graphs with disconnected components as well. HiGen learns to reverse a graph coarsening algorithm such as Louvain, and if a graph is disconnected, the output of the coarsening algorithm is an $\mathcal{HG}$ which has a disconnected graph at the top level ($l=1$ in figure (1.a)). To illustrate, let's consider the example graph in Figure 1.a being disconnected into two left and right components (by removing the edge between the yellow and blue communities and the edge between the cyan and orange communities, and subsequently removing their corresponding edge with $weight=1$ at the top level ($l=1$)), therefore the coarsened graph at the top level is a disconnected community composed of two components. Consequently, in a hierarchical graph generation, HiGen first generates a community with two disconnected components for the $l=1$ then the subsequent final graph (leaf level graph) will be a disconnected graph.
> An advantageous feature of the proposed model is its ability to enforce connected graph generation by constraining the total sum of the candidate edge weights to $v_t >0$ in the recursive *stick-breaking* process. This was achieved by modeling and sampling an auxiliary variable $v’_t  >= 0 $ from the binomial distribution in theorem 3.2 and then setting its effective value as $v_t = v’_t +1$. This guarantees that there is at least one edge between the new node and the already generated community in the autoregressive community generation process. As our experimental datasets had connected graphs, we modeled them as connected using the aforementioned technique. However, to potentially generate disconnected graphs, we need to relax the community generation at level $l=1$ to include disconnected communities while we can restrict to connected community generation for the rest of the levels, $l>1$.
>
> 2. **Capturing and maintaining of global graph structure:** Indeed, a key advantage of the proposed hierarchical method is its ability to condition the graph generation at each level on the graph at the parent level, as modeled by $p(\mathcal{G}^l | \mathcal{G}^{l-1})$. Therefore, when generating community graphs at level $l$, the previously generated coarser graph at its parent level inherently encapsulates the global structure of the graph, therefore concatenating its node/edge embedding in community and bipartite generation (eq (6, 7) ) effectively captures global structure and  long-range interactions within the graphs. Therefore, by looking at its parent graph, HiGen ensures maintaining control over the global graph distribution while the non hierarchical methods lack such capability. This mechanism enables HiGen to take advantage of consistency across multiple levels of global graph abstraction throughout the hierarchical generation process.
>
> I hope these have addressed all your questions. We highly value your insights and suggestions and are committed to addressing any outstanding concern you may have.

---

> > ### Comment · Reviewer_D2iQ · 2023-11-20
> > **Thank you**
> >
> > Thank you for your thorough response; my concerns have been effectively addressed. I find the idea intriguing and believe it has practical applications in various real-world scenarios. Therefore, I keep the recommendation of this paper.

---

### Official Review · Reviewer_fvzA · 2023-11-02

**Soundness:** 4 excellent
**Presentation:** 4 excellent
**Contribution:** 2 fair
**Rating:** 8
**Confidence:** 3

**Summary:**

The authors mainly aim to propose a graph generative model that can capture hierarchical structures. To this end, the authors propose a coarse-to-fine manner method that generate the graph structures by modeling the distribution of connectivity as a recursive multinomial distribution and decomposing the graph generation process into the generation of communities and bipartites at each level. The proposed method is evaluated on the general graph generation task and 3D point cloud generation tasks.

**Strengths:**

* The proposed coarse-to-fine manner is an effective method to generate larger graphs as it can gradually recover the structure with the knowledge of the hierarchical clusters of the graphs.
* The performances of the proposed method are superior to the existing autoregressive and diffusion models.
* The authors provide an informative ablation study on the effect of the node ordering and the graph partitioning method.

**Weaknesses:**

* The proposed method is effective in generating the graph structures. However, I am concerned that it requires an additional generator to generate the graph attributes for the realistic graph generation and it could be a harder problem to generate the graph attributes correctly only given the structures.
* To strengthen the contribution of the proposed method, it would be better to evaluate it on the molecular graphs.

**Questions:**

* How does the performance change depending on the partitioning algorithm?
* Could you elaborate why HiGen outperforms HiGen-m?

---

> ### Author Response · Authors · 2023-11-16
>
> Dear reviewer,
>
> We appreciate your comprehensive review and the insightful feedback. In the following, we answer the questions and provide clarification on the concerns:
>
> **Attributed Graph:** HiGen offers a versatile framework for efficient and scalable graph generation,  emphasizing the capture of hierarchical and community structures in graphs. Adapting HiGen to handle attributed graphs involves reversing the partitioning (coarsening) algorithms tailored for clustering attributed graphs like *molecular structures*, for example a GNN-based clustering method inspired by the spectral relaxation of modularity metric for graphs with node attributes proposed by Tsitsulin et al., (2020). Toward that goal, we need to modify the proposed community generation and cross-community predictor to learn attributed edges or use the graph generation models with such capability.
>
> Extending HiGen for graphs with edge types involves assigning a weight vector $\mathbf{w}\_{i, j}^l \in \mathbb{Z}^d$ to each edge $e{i, j}^l = (i, j)$, where $d$ is the number of edge types and each feature indicates the number of one specific edge type that the edge $e_{i,j}^l$ at a level $l$ is representing. The autoregressive multinomial and binomial generative models presented in this work can then be applied to each dimension so that the attributed graph at level $l+1$ is generated based on its parent attributed graph $\mathcal{G}^l$. Note that, in this approach, the attribute of node $v_i$ can be added as a self loop edge $e_{i,i} = (i, i)$ with weight $\mathbf{w}\_{i,i}=x_i$. Consequently,  the model only needs to predict the edge attributes (or edge types) , aligning with HiGen's edge weight generation capabilities.
>
> Another approach, as you mentioned, is training an additional predictor, such as a GNN model for edge/node classification, dedicated to predicting edge and node types based on the graph structure. This model doesn’t have the difficulties associated with graph topology generation – such as graph isomorphism, automorphism and edge independence – focusing solely on predicting edge/node attributes. This additional model can be tailored to specific application requirements and characteristics.
> However, the primary focus of this work is to demonstrate the efficiency and scalability of the hierarchical framework. Addressing attributed graphs is acknowledged as a potential future avenue for research.
>
> - **Performance Dependence on Partitioning Algorithm:** In Table 12 (Appendix E), the performance of the proposed model is compared for two families of graph partitioning algorithms:  modularity based Louvain algorithm and spectral clustering based algorithm. These results demonstrate the robustness of HiGen against different graph partitioning functions.  Furthermore, in Table 2, the performance of HiGen is provided for the large point cloud dataset against two different hyperparameters of graph coarsening algorithm, $L=2$ vs $L=3$.
>
>
> - **Comparison of HiGen and HiGen-m:** The performance difference between HiGen and HiGen-m is primarily attributed to the modeling choices for the probability distribution of community edges' weights at the leaf level. HiGen employs a mixture of Bernoulli distributions for this purpose, while HiGen-m opts for a mixture of multinomial distributions. The key distinction lies in the nature of the edge weights at the leaf level in our experiments, where the distribution of binary edge weights at the leaf level can be better specified by a mixture of Bernoulli. This can explain why HiGen outperforms HiGen-m in most metrics.

---

### Author Response · Authors · 2023-11-20
**Manuscript update summary**

Dear Reviewers,

Thank you for your valuable feedback. We have carefully considered your comments and incorporated some  discussion into the manuscripts. Here is a summary of the key revisions:
1. The training and sampling algorithms have been elaborated in detail in Appendix C.
2. A detailed complexity analysis and comparison  are added in Appendix C.
3. Additional discussions on connected graph generation and the generation of attributed graphs, such as molecular graphs, have been included in the Appendix.
4. Typos identified have been corrected.

We appreciate your time and insights and remain open to any further concerns you may have.

---

### Meta-Review · Area_Chair_TQRY · 2023-12-05

**Metareview:**

The paper introduces a new method to generate synthetic graphs with good hierarchical structure. The main idea behind the new method is to design a coarse-to-fine approach to generate graph substructure.

The proposed method is elegant and very scalable and has very good experimental results. The paper is also very clear and easy-to-read. Overall, the paper is a nice contribution and it would be nice to have it in ICLR.

**Justification For Why Not Higher Score:**

The results are nice, but the problem is a bit niche so it is a good fit for a poster.

**Justification For Why Not Lower Score:**

The paper presents a new efficient and very effective algorithm for an interesting problem(even if the problem is a bit niche)

---

### Decision · Program_Chairs · 2024-01-16

Accept (poster)